# Towards Stable Backdoor Purification through Feature Shift Tuning

**Rui Min[1*], Zeyu Qin[1*], Li Shen[2], Minhao Cheng[1]**
[1]Department of Computer Science & Engineering, HKUST
[2]JD Explore Academy
{rminaa, zeyu.qin}@connect.ust.hk
mathshenli@gmail.com
minhaocheng@ust.hk

## Abstract

It has been widely observed that deep neural networks (DNN) are vulnerable to backdoor attacks where attackers could manipulate the model behavior maliciously by tampering with a small set of training samples. Although a line of defense methods is proposed to mitigate this threat, they either require complicated modifications to the training process or heavily rely on the specific model architecture, which makes them hard to deploy into real-world applications. Therefore, in this paper, we instead start with fine-tuning, one of the most common and easy-to-deploy backdoor defenses, through comprehensive evaluations against diverse attack scenarios. Observations made through initial experiments show that in contrast to the promising defensive results on high poisoning rates, vanilla tuning methods completely fail at low poisoning rate scenarios. Our analysis shows that with the low poisoning rate, the entanglement between backdoor and clean features undermines the effect of tuning-based defenses. Therefore, it is necessary to disentangle the backdoor and clean features in order to improve backdoor purification. To address this, we introduce Feature Shift Tuning (FST), a method for tuning-based backdoor purification. Specifically, FST encourages feature shifts by actively deviating the classifier weights from the originally compromised weights. Extensive experiments demonstrate that our FST provides consistently stable performance under different attack settings. Without complex parameter adjustments, FST also achieves much lower tuning costs, only 10 epochs. Our codes are available at `https://github.com/AISafety-HKUST/stable_backdoor_purification`.

## 1 Introduction

Deep Neural Networks (DNNs) are shown vulnerable to various security threats. One of the main security issues is backdoor attack [6, 9, 10] that inserts malicious backdoors into DNNs by manipulating the training data or controlling the training process.

To alleviate backdoor threats, many defense methods [36] have been proposed, such as robust training [13, 17, 38] and post-processing purification methods [20, 37, 41]. However, robust training methods require complex modifications to model training process [13, 38], resulting in substantially increased training costs, particularly for large models. Pruning-based purification methods are sensitive to hyperparameters and model architecture [37, 41], which makes them hard to deploy in real-world applications.

---

[*]Equal contribution. Email to `zeyu.qin@connect.ust.hk`

37th Conference on Neural Information Processing Systems (NeurIPS 2023).

Fine-tuning (FT) methods have been adopted to improve models' robustness against backdoor attacks [14, 20, 36] since they can be easily combined with existing training methods and various model architectures. Additionally, FT methods require less computational resources, making them one of the most popular transfer learning paradigms for large pretrained models [5, 21, 25, 28, 35, 40]. However, FT methods have not been sufficiently evaluated under various attack settings, particularly in the more practical low poisoning rate regime [2, 4]. Therefore, we begin by conducting a thorough assessment of widely-used tuning methods, vanilla FT and simple Linear Probing (LP), under different attack configurations.

In this work, we focus on *data-poisoning backdoor attacks*, as they efficiently exploit security risks in more practical settings [3, 4, 9, 27]. We start our study on *whether these commonly used FT methods can efficiently and consistently purify backdoor triggers in diverse scenarios.* We observe that *vanilla FT and LP can not achieve stable robustness against backdoor attacks while maintaining clean accuracy*. What's worse, although they show promising results under high poisoning rates (20%, 10%), they completely fail under low poisoning rates (5%, 1%, 0.5%). As shown in Figure 3, we investigate this failure mode and find that model's learned features (representations before linear classifier) have a significant difference under different poisoning rates, especially in terms of separability between clean features and backdoor features. For low poisoning rate scenarios, clean and backdoor features are tangled together so that the simple LP is not sufficient for breaking mapping between input triggers and targeted label without feature shifts. Inspired by these findings, we first try two simple strategies (shown in Figure 1), *FE-tuning* and *FT-init* based on LP and FT, respectively, to encourage shifts on learned features. In contrast to LP, FE-tuning only tunes the feature extractor with the frozen and re-initialized linear classifier. FT-init first randomly initializes the linear classifier and then conducts end-to-end tuning. Experimental results show that these two methods, especially FE-tuning, improve backdoor robustness for low poisoning rates, which confirms the importance of promoting shifts in learned features.

Though these two initial methods can boost backdoor robustness, they still face an unsatisfactory trade-off between defense performance and clean accuracy. With analysis of the mechanism behind those two simple strategies, we further proposed a stronger defense method, *Feature Shift Tuning* (FST) (shown in Figure 1). Based on the original classification loss, FST contains an extra penalty, $\langle \boldsymbol{w}, \boldsymbol{w}^{ori} \rangle$, the inner product between the tuned classifier weight $\boldsymbol{w}$ and the original backdoored classifier weight $\boldsymbol{w}^{ori}$. During the end-to-end tuning process, FST can actively shift backdoor features by encouraging the difference between $\boldsymbol{w}$ and $\boldsymbol{w}^{ori}$ (shown in Figure 3). Extensive experiments have demonstrated that FST achieves better and more stable defense performance across various attack settings with maintaining clean accuracy compared with existing defense methods. Our method also significantly improves efficiency with fewer tuning costs compared with other tuning methods, which makes it a more convenient option for practical applications. To summarize, our contributions are:

- We conduct extensive evaluations on various tuning strategies and find that while vanilla Fine-tuning (FT) and simple Linear Probing (LP) exhibit promising results in high poisoning rate scenarios, they fail completely in low poisoning rate scenarios.

- We investigate the failure mode and discover that the reason behind this lies in varying levels of entanglement between clean and backdoor features across different poisoning rates. We further propose two initial methods to verify our analysis.

- Based on our initial experiments, we propose Feature Shift Tuning (FST). FST aims to enhance backdoor purification by encouraging feature shifts that increase the separability between clean and backdoor features. This is achieved by actively deviating the tuned classifier weight from its originally compromised weight.

- Extensive experiments show that FST outperforms existing backdoor defense and other tuning methods. This demonstrates its superior and more stable defense performance against various poisoning-based backdoor attacks while maintaining accuracy and efficiency.

## 2 Background and related work

**Backdoor Attacks.** Backdoor attacks aim to mislead the backdoored model to exhibit abnormal behavior on samples stamped with the backdoor trigger but behave normally on all benign samples. They can be classified into 2 categories [36]: **(1)** data-poisoning attacks: the attacker inserts a backdoor trigger into the model by manipulating the training sample $(\boldsymbol{x}, \boldsymbol{y}) \in (\mathcal{X}, \mathcal{Y})$, like adding a

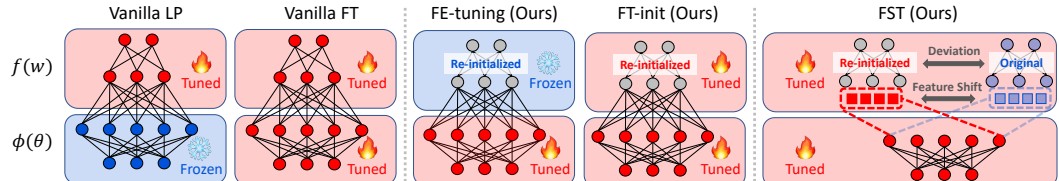

Figure 1: The first two methods, LP and vanilla FT, are adopted in Section 3.1. The middle two methods, FE-tuning and FT-init, are proposed in Section 3.2. The final one, FST, is our method introduced in Section 4.

small patch in clean image $x$ and assign the corresponding class $y$ to an attacker-designated target label $y_t$. [3, 6, 9, 10, 18, 33]; **(2)** training-control attacks: the attacker can control both the training process and training data simultaneously [23, 24]. With fewer assumptions about attackers' capability, data-poisoning attacks are much more practical in real-world scenarios [4, 9, 29] and have led to increasingly serious security risks [3, 27]. Therefore, in this work, we mainly focus on data-poisoning backdoor attacks.

**Backdoor Defense.** Existing backdoor defense strategies could be roughly categorized into robust training [13, 17, 38] and post-processing purification methods [20, 37, 41]. Robust training aims to prevent learning backdoored triggers during the training phase. However, their methods suffer from accuracy degradation and significantly increase the model training costs [17, 38], which is impractical for large-scale model training. Post-processing purification instead aims to remove the potential backdoor features in a well-trained model. The defender first identifies the compromised neurons and then prunes or unlearns them [20, 34, 37, 41, 42]. However, pruning-based and unlearning methods also sacrifice clean accuracy and lack generalization across different model architectures.

**Preliminaries of backdoor fine-tuning methods.** Without crafting sophisticated defense strategies, recent studies [14, 20, 27, 42] propose defense strategies based on simple Fine-tuning. Here, we introduce two widely used fine-tuning paradigms namely vanilla Fine-tuning (FT) and Linear Probing (LP) (shown in Figure 1) since they would serve as two strong baselines in our following sections. For each tuned model, we denote the feature extractor as $\phi(\theta; x) : \mathcal{X} \to \phi(x)$ and linear classifier as $f(w; x) = w^T \phi(x) : \phi(x) \to \mathcal{Y}$. To implement the fine-tuning, both tuning strategies need a set of training samples denoted as $\mathcal{D}_T \subset (\mathcal{X}, \mathcal{Y})$ to update the model parameters while focusing on different parameter space. Following previous works [5, 14, 16, 20, 21], we implement vanilla FT by updating the whole parameters $\{\theta, w\}$; regarding the LP, we only tunes the linear classifier $f(w)$ without modification on the frozen $\phi(\theta)$.

**Evaluation Metrics.** We take two evaluation metrics, including *Clean Accuracy (C-Acc)* (i.e., the prediction accuracy of clean samples) and *Attack Success Rate (ASR)* (i.e., the prediction accuracy of poisoned samples to the target class). A lower ASR indicates a better defense performance.

# 3 Revisiting Backdoor Robustness of Fine-tuning Methods

In this section, we evaluate the aforementioned two widely used fine-tuning paradigms' defensive performance against backdoor attacks with various poisoning rates (FT and LP). Despite that the vanilla FT has been adopted in previous defense work [14, 20, 36, 42], it has not been sufficiently evaluated under various attack settings, particularly in more practical low poisoning rate scenarios. The simple LP method is widely adopted in transfer learning works [5, 16, 21, 22, 28] but still rarely tested for improving model robustness against backdoor attacks. For FT, we try various learning rates during tuning: $0.01, 0.02$, and $0.03$. For LP, following [16], we try larger learning rates: $0.3, 0.5$, and $0.7$. We also demonstrate these two methods in Figure 1.

We conduct evaluations on widely used CIFAR-10 [15] dataset with ResNet-18 [11] and test 4 representative data-poisoning attacks including BadNet [10], Blended [6], SSBA [18] and Label-Consistent attack (LC) [33]. We include various poisoning rates, $20\%, 10\%, 5\%, 1\%$, and $0.5\%$ for attacks except for LC since the maximum poisoning rate for LC on CIFAR-10 is $10\%$. Following previous work [36], we set the target label $y_t$ as class $0$. Additional results on other models and datasets are shown in *Appendix C.1*. We aim to figure out *whether these commonly used FT methods can efficiently and consistently purify backdoor triggers in various attack settings.*

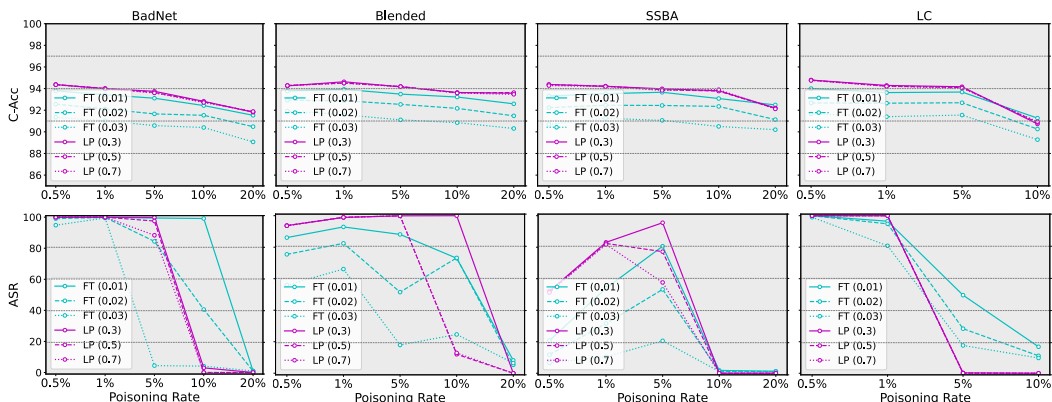

Figure 2: The clean accuracy and ASR of 4 attacks on CIFAR-10 and ResNet-18. The $x$-axis means the poisoning rates. The blue and purple lines represent FT and LP, respectively. For FT, we try various learning rates during tuning: $0.01, 0.02$, and $0.03$. For LP, we try learning rates: $0.3, 0.5$, and $0.7$.

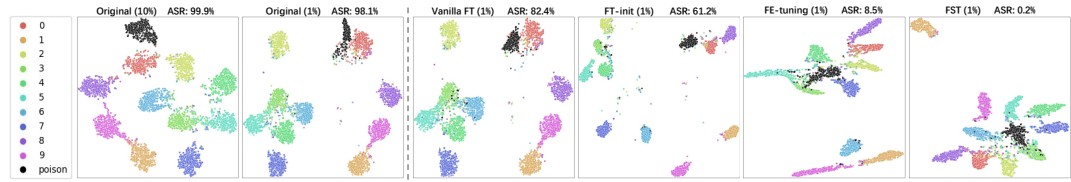

Figure 3: We take T-SNE visualizations on features from feature extractors with *Blended attack*. We adopt half of the CIFAR-10 test set and ResNet-18. Each color denotes each class, and **Black** points represent backdoored samples. The targeted class is **0 (Red)**. **(1)** The first two figures are feature visualizations of original backdoored models with $10\%$ and $1\%$ poisoning rates; **(2)** The rest 4 figures are feature visualizations after using different tuning methods for $1\%$ poisoning rate. We also add the corresponding ASR.

## 3.1 Revisiting Fine-tuning under Various Poisoning Rates

We stress that the defense method should effectively purify the backdoor triggers while maintaining good clean accuracy. Therefore, we mainly focus on defense performance with a satisfactory clean accuracy level (around $92\%$). The results are shown in Figure 2.

*Vanilla FT and LP can purify backdoored models for high poisoning rates.* From Figure 2, We observe that for high poisoning rates ($\geq 10\%$ for BadNet, Blended, and SSBA; $5\%$ for LC), both vanilla FT and LP can effectively mitigate backdoor attacks. Specifically, LP (purple lines) significantly reduces the ASR below $5\%$ on all attacks without significant accuracy degradation ($\leq 2.5\%$). Compared with vanilla FT, by simply tuning the linear classifier, LP can achieve better robustness and clean accuracy.

*Vanilla FT and LP both fail to purify backdoor triggers for low poisoning rates.* In contrast to their promising performance under high poisoning rates, both tuning methods fail to defend against backdoor attacks with low poisoning rates ($\leq 5\%$ for BadNet, Blended, and SSBA; $< 5\%$ for LC). Vanilla FT with larger learning rates performs slightly better on Blended attacks, but it also sacrifices clean accuracy, leading to an intolerant clean accuracy drop. The only exception is related to the SSBA results, where the ASR after tuning at a $0.5\%$ poisoning rate is low. This can be attributed to the fact that the original backdoored models have a relatively low ASR.

## 3.2 Exploration of Unstable Defense Performance of Fine-tuning Methods

From the results, a question then comes out: **What leads to differences in defense performance of tuning methods under various poisoning rate settings?** We believe the potential reason for this phenomenon is that *the learned features from feature extractors of backdoored models differ at different poisoning rates, especially in terms of the separability between clean features and backdoor features.* We conduct T-SNE visualizations on learned features from backdoored models with Blended attack ($10\%$ and $1\%$ poisoning rates). The results are shown in Figure 3. The targeted class samples are marked with Red color and Black points are backdoored samples. As shown in first two figures,

Table 1: Purification performance of fine-tuning against four types of backdoor attacks with low poisoning rates. All the metrics are measured in percentage (%).

| Attack | Poisoning rate | No defense | | LP | | FE-tuning | | Vanilla FT | | FT-init | |
|---|---|---|---|---|---|---|---|---|---|---|---|
| | | C-Acc(↑) | ASR(↓) | C-Acc(↑) | ASR(↓) | C-Acc(↑) | ASR(↓) | C-Acc(↑) | ASR(↓) | C-Acc(↑) | ASR(↓) |
| BadNet | 1% | 94.52 | 100 | **94.02** | 100 | 91.56 | **3.18** | 92.12 | 99.83 | 93.37 | 16.72 |
| | 0.5% | 94.79 | 100 | **94.37** | 100 | 92.37 | **7.41** | 92.56 | 99.07 | 93.90 | 79.32 |
| Blended | 1% | 95.13 | 98.12 | **94.51** | 98.98 | 92.03 | **8.50** | 92.97 | 82.36 | 93.88 | 61.23 |
| | 0.5% | 94.45 | 92.46 | **94.29** | 93.72 | 91.84 | **6.48** | 92.95 | 75.99 | 93.71 | 49.80 |
| SSBA | 1% | 94.83 | 79.54 | **94.23** | 82.39 | 91.99 | **5.58** | 92.59 | 30.16 | 93.51 | 21.04 |
| | 0.5% | 94.50 | 50.20 | **94.37** | 51.67 | 91.31 | **2.50** | 92.36 | 12.69 | 93.41 | 6.69 |
| LC | 1% | 94.33 | 99.16 | **94.23** | 99.98 | 91.70 | 64.97 | 92.65 | 94.83 | 93.54 | 89.86 |
| | 0.5% | 94.89 | 100 | **94.78** | 100 | 91.65 | **22.22** | 92.67 | 99.96 | 93.83 | 96.16 |
| Average | | 94.68 | 89.94 | **94.35** | 90.91 | 91.81 | **15.11** | 92.61 | 74.36 | 93.64 | 52.60 |

under high poisoning rate (10%), backdoor features (black points) are clearly separable from clean features (red points) and thus could be easily purified by only tuning the $f(w)$; however for low poisoning rate (1%), clean and backdoor features are tangled together so that the simple LP is not sufficient for breaking mapping between input triggers and targeted label without further feature shifts. Furthermore, as depicted in the third figure of Figure 3, though vanilla FT updates the whole network containing both $\theta$ and $w$, it still suffers from providing insufficient feature modification, leading to the failure of backdoor defense.

**Can Feature Shift Help Enhance the Purification Performance of Fine-tuning?** Based on our analysis, we start with a simple strategy in that we could improve backdoor robustness for low poisoning rates by encouraging shifts in learned features. We then propose separate solutions for both LP and vanilla FT to evaluate the effect of feature shift as well as the robustness against two low poisoning rates, 1% and 0.5%. As shown in Table 1, specifically, for the LP, we freeze $f(w)$ and only tune $\phi(\theta)$. However, our initial experiments show that directly tuning $\phi(\theta)$ does not sufficiently modify learned features since the fixed backdoor linear classifier (denoted as $f(w^{ori})$) may still restrict modifications of learned features. Inspired by the previous work [27], we first randomly re-initialize the linear classifier and then tune only $\phi(\theta)$ (denoted as FE-tuning). For vanilla FT, we also fine-tune the whole network with a randomly re-initialized linear classifier (denoted as FT-init). More implementation details of FE-tuning and FT-init are shown in *Appendix* B.3.

**Evaluations Verify That Encouraging Feature Shift Could Help Purify Backdoored Models.** We observe that both FE-tuning and FT-init could significantly enhance the performance of backdoor purification compared to previous fine-tuning with an average drop of 77.70% and 35.61% on ASR respectively. Specifically, FE-tuning leads to a much larger improvement, an ASR decrease of over 74%. As shown in the fifth figure of Figure 3, *after FE-tuning, backdoor features could be clearly separated from the clean features of the target class (red).* However, this simple strategy also leads to a decrease in clean accuracy (around 2.9%) in comparison to the original LP, due to the totally randomized classifier. While for the FT-init, the robustness improvements are less significant. As shown in the fourth figure of Figure 3, simply fine-tuning the backdoor model with re-initialized $f(w)$ can not lead to enough shifts on backdoor features under the low poisoning rate. The backdoor and clean features of the target class still remain tangled, similar to the original and vanilla FT.

In summary, the results of these two initial methods confirm the fact that *encouraging shifts on learned features is an effective method for purifying backdoors at low poisoning rates.* However, both methods still experience a serious clean accuracy drop or fail to achieve satisfactory improvements in robustness. To further enhance the purification performance, we propose a stronger tuning defense in Section 4 that addresses both issues in a unified manner.

## 4 Feature Shift Tuning: Unified Method to Achieve Stable Improvements

Based on our initial findings, we propose a stronger tuning defense paradigm called **Feature Shift Tuning (FST)**. FST is an end-to-end tuning method and actively shifts features by encouraging the discrepancy between the tuned classifier weight $w$ and the original backdoored classifier weight $w^{ori}$. The illustration of FST is shown in Figure 1. Formally, starting with reinitializing the linear classifier

$\boldsymbol{w}$, our method aims to solve the following optimization problem:

$$\min_{\boldsymbol{\theta}, \boldsymbol{w}} \left\{ \mathbb{E}_{(\boldsymbol{x}, \boldsymbol{y}) \sim \mathcal{D}_T}[\mathcal{L}(\boldsymbol{f}(\boldsymbol{w}; \boldsymbol{\phi}(\boldsymbol{\theta}; \boldsymbol{x})), \boldsymbol{y})] + \alpha \left\langle \boldsymbol{w}, \boldsymbol{w}^{ori} \right\rangle, \quad s.t. \|\boldsymbol{w}\|_2 = C \right\} \tag{1}$$

where $\mathcal{L}$ stands for the original classification cross-entropy loss over whole model parameters to maintain clean accuracy and $\langle \cdot, \cdot \rangle$ denotes the inner product. By adding the regularization on the $\left\langle \boldsymbol{w}, \boldsymbol{w}^{ori} \right\rangle$, FST encourages discrepancy between $\boldsymbol{w}$ and $\boldsymbol{w}^{ori}$ to guide more shifts on learned features of $\boldsymbol{\phi}(\boldsymbol{\theta})$. The $\alpha$ balances these two loss terms, the trade-off between clean accuracy and backdoor robustness from feature shift. While maintaining clean accuracy, a larger $\alpha$ would bring more feature shifts and better backdoor robustness. We simply choose the inner-product $\left\langle \boldsymbol{w}, \boldsymbol{w}^{ori} \right\rangle$ to measure the difference between weights of linear classifiers, since it is easy to implement and could provide satisfactory defensive performance in our initial experiments. Other metrics for measuring differences can also be explored in the future.

**Projection Constraint.** To avoid the $\boldsymbol{w}$ exploding and the inner product dominating the loss function during the fine-tuning process, we add an extra constraint on the norm of $\boldsymbol{w}$ to stabilize the tuning process. To reduce tuning cost, we directly set it as $\|\boldsymbol{w}^{ori}\|$ instead of manually tuning it. *With the constraint to shrink the feasible set, our method quickly converges in just a few epochs while achieving significant robustness improvement.* Compared with previous tuning methods, our method further enhances robustness against backdoor attacks and also greatly improves tuning efficiency (Shown in Section 5.3). Shrinking the range of feasible set also brings extra benefits. *It significantly reduces the requirement for clean samples during the FST process.* As shown in Figure 7 (c,d), FST consistently performs well across various tuning data sizes, even when the tuning set only contains 50 samples. The ablation study of

---

**Algorithm 1** Feature Shift Tuning (FST)

**Input:** Tuning dataset $\mathcal{D}_T = \{\boldsymbol{x}_i, \boldsymbol{y}_i\}_{i=1}^{N}$; backdoored model $\boldsymbol{f}(\boldsymbol{w}^{ori}; \boldsymbol{\phi}(\boldsymbol{\theta}))$; learning rate $\eta$; factor $\alpha$; tuning iterations $I$

**Output:** Purified model

1: Initialize $\boldsymbol{w}^0$ with random weights
2: Initialize $\boldsymbol{\theta}^0$ with $\boldsymbol{\theta}$
3: **for** $i = 0, 1, \dots, I$ **do**
4:     Sample mini-batch $\mathcal{B}_i$ from tuning set $\mathcal{D}_T$
5:     Calculate gradients of $\boldsymbol{\theta}^i$ and $\boldsymbol{w}^i$:
    $\boldsymbol{g}_\theta^i = \nabla_{\boldsymbol{\theta}^i} \frac{1}{|\mathcal{B}_i|} \sum_{\boldsymbol{x} \in \mathcal{B}_i} \mathcal{L}(\boldsymbol{f}(\boldsymbol{w}^i; \boldsymbol{\phi}(\boldsymbol{\theta}^i; \boldsymbol{x})), \boldsymbol{y})$;
    $\boldsymbol{g}_w^i = \nabla_{\boldsymbol{w}^i} [\frac{1}{|\mathcal{B}_i|} \sum_{\boldsymbol{x} \in \mathcal{B}_i} \mathcal{L}(\boldsymbol{f}(\boldsymbol{w}^i; \boldsymbol{\phi}(\boldsymbol{\theta}^i; \boldsymbol{x})), \boldsymbol{y}) + \alpha \left\langle \boldsymbol{w}^i, \boldsymbol{w}^{ori} \right\rangle]$;
6:     Update model parameters $\boldsymbol{\theta}^{i+1} = \boldsymbol{\theta}^i - \eta \boldsymbol{g}_\theta^i$, $\boldsymbol{w}^{i+1} = \boldsymbol{w}^i - \eta \boldsymbol{g}_w^i$
7:     Norm projection of the linear classifier $\boldsymbol{w}^{i+1} = \frac{\boldsymbol{w}^{i+1}}{\|\boldsymbol{w}^{i+1}\|_2} \|\boldsymbol{w}^{ori}\|_2$
8: **end for**
9: **return** Purified model $\boldsymbol{f}(\boldsymbol{w}^I; \boldsymbol{\phi}(\boldsymbol{\theta}^I))$

---

$\alpha$ is also provided in Section 5.3 and showcases that our method is not sensitive to the selection of $\alpha$ in a wide range. The overall optimization procedure is summarized in Algorithm 1.

**Unified Improvement for Previous Tuning Methods.** We discuss the connections between FST and our previous FE-tuning and FT-init to interpret why FST could achieve unified improvements on backdoor robustness and clean accuracy. Our objective function Eq.1 could be decomposed into three parts, by minimizing $\mathbb{E}_{(\boldsymbol{x}, \boldsymbol{y}) \sim \mathcal{D}_T} \mathcal{L}(\boldsymbol{f}(\boldsymbol{w}; \boldsymbol{\phi}(\boldsymbol{\theta}; \boldsymbol{x})), \boldsymbol{y})$ over $\boldsymbol{w}$ and $\boldsymbol{\theta}$ respectively, and $\alpha \left\langle \boldsymbol{w}, \boldsymbol{w}^{ori} \right\rangle$ over $\boldsymbol{w}$. Compared with FE-tuning, FST brings an extra loss term on $\min_{\boldsymbol{w}} \mathbb{E}_{(\boldsymbol{x}, \boldsymbol{y}) \sim \mathcal{D}_T} \mathcal{L}(\boldsymbol{f}(\boldsymbol{w}; \boldsymbol{\phi}(\boldsymbol{\theta}; \boldsymbol{x})), \boldsymbol{y}) + \alpha \left\langle \boldsymbol{w}, \boldsymbol{w}^{ori} \right\rangle$ to update linear classifier $\boldsymbol{w}$. In other words, while encouraging feature shifts, FST updates the linear classifier with the original loss term to improve the models' clean performance. Compared with the FT-init, by adopting $\alpha \left\langle \boldsymbol{w}, \boldsymbol{w}^{ori} \right\rangle$, FST encourages discrepancy between tuned $\boldsymbol{w}$ and original $\boldsymbol{w}^{ori}$ to guide more shifts on learned features.

**Weights of Linear Classifier Could Be Good Proxy of Learned Features.** Here, we discuss why we choose the discrepancy between linear classifiers as our regularization in FST. Recalling that our goal is to introduce more shifts in learned features, especially for backdoor features. Therefore, the most direct approach is to explicitly increase the difference between backdoor and clean feature distributions. However, we can not obtain backdoor features without inputting backdoor triggers. We consider that *weights of the original compromised linear classifier could be a good proxy of learned features*. Therefore, we encourage discrepancy between tuned $\boldsymbol{w}$ and original $\boldsymbol{w}^{ori}$ rather than trying to explicitly promote discrepancy between feature distributions.

# 5 Experiments

## 5.1 Experimental Settings

**Datasets and Models.** We conduct experiments on four widely used image classification datasets, CIFAR-10 [15], GTSRB [32], Tiny-ImageNet [8], and CIFAR-100 [15]. Following previous works [13, 20, 24, 36], we implement backdoor attacks on ResNet-18 [11] for both CIFAR-10 and GTSRB and also explore other architectures in Section 5.3. For CIFAR-100 and Tiny-ImageNet, we adopt pre-trained SwinTransformer [21] (Swin). For both the CIFAR-10 and GTSRB, we follow the previous work [41] and leave 2% of original training data as the tuning dataset. For the CIFAR-100 and Tiny-ImageNet, we note that a small tuning dataset would hurt the model performance and therefore we increase the tuning dataset to 5% of the training set.

**Attack Settings.** All backdoor attacks are implemented with BackdoorBench [1]. We conduct evaluations against 6 representative data-poisoning backdoors, including 4 dirty-label attacks (BadNet [10], Blended attack [6], WaNet [24], SSBA[18]) and 2 clean-label attacks (SIG attack [1], and Label-consistent attack (LC) [33]). For all the tasks, we set the target label $y_t$ to 0, and focus on low poisoning rates, 5%, 1%, and 0.5% in the main experimental section. We also contain experiments of **high poisoning rates**, 10%, 20%, and 30%, in *Appendix* C.2. For the GTSRB dataset, we do not include LC since it can not insert backdoors into models (ASR < 10%). Out of all the attacks attempted on Swin and Tiny-ImageNet, only BadNet, Blended, and SSBA were able to successfully insert backdoor triggers into models at low poisoning rates. Other attacks resulted in an ASR of less than 20%. Therefore, we only show the evaluations of these three attacks. More details about attack settings and trigger demonstrations are shown in *Appendix* B.2.

**Baseline Defense Settings.** We compare our FST with 4 tuning-based defenses including Fine-tuning with Sharpness-Aware Minimization (FT+SAM), a simplified version adopted from [42], Natural Gradient Fine-tuning (NGF) [14], FE-tuning and FT-init proposed in Section 3.2. We also take a comparison with 2 extra strategies including Adversarial Neural Pruning (ANP) [37] and Implicit Backdoor Adversarial Unlearning (I-BAU) [38] which achieve outstanding performance in BackdoorBench [37]. The implementation details of baseline methods are shown in the *Appendix* B.3. For our FST, we adopt SGD with an initial learning rate of 0.01 and set the momentum as 0.9 for both CIFAR-10 and GTSRB datasets and decrease the learning rate to 0.001 for both CIFAR-100 and Tiny-ImageNet datasets to prevent the large degradation of the original performance. We fine-tune the models for 10 epochs on the CIFAR-10; 15 epochs on the GTSRB, CIFAR-100 and Tiny-ImageNet. We set the $\alpha$ as 0.2 for CIFAR-10; 0.1 for GTSRB; 0.001 for both the CIFAR-100 and Tiny-ImageNet.

## 5.2 Defense Performance against Backdoor Attacks

In this section, we show the performance comparison between FST with tuning-based backdoor defenses (FT+SAM [42], NGF [14], FE-tuning and FT-init) and current state-of-the-art defense methods, ANP [37] and I-BAU [38]. We demonstrate results of CIFAR-10, GTSRB, and Tiny-ImageNet on Table 2, 3, and 4, repectively. We leave results on CIFAR-100 to *Appendix* C.3.

Experimental results show that *our proposed FST achieves superior backdoor purification performance compared with existing defense methods*. Apart from Tiny-ImageNet, FST achieves the best performances on CIFAR-10 and GTSRB. The average ASR across all attacks on three datasets are below 1%, 0.52% in CIFAR-10, 0.41% in GTSRB, and 0.19% in Tiny-ImageNet, respectively. Regarding two tuning defense methods, FT+SAM and NGF, FST significantly improves backdoor robustness with larger ASR average drops on three datasets by 34.04% and 26.91%. Compared with state-of-the-art methods, ANP and I-BAU, on CIFAR-10 and GTSRB, our method outperforms with a large margin by 11.34% and 32.16% on average ASR, respectively. ANP is only conducted in BatchNorm layers of ConvNets in source codes. Therefore, it can not be directly conducted in SwinTransformer. Worth noting is that *our method achieves the most stable defense performance across various attack settings* with 0.68% on the average standard deviation for ASR. At the same time, our method still maintains high clean accuracy with 93.07% on CIFAR-10, 96.17% on GTSRB, and 79.67% on Tiny-ImageNet on average. Compared to ANP and I-BAU, FST not only enhances backdoor robustness but also improves clean accuracy with a significant boost on the clean accuracy by 2.5% and 1.78%, respectively.

---

[1] https://github.com/SCLBD/backdoorbench

Table 2: Defense results under various poisoning rates. The experiments are conducted on the CIFAR-10 dataset with ResNet-18. All the metrics are measured in percentage (%). The best results are bold.

| Attack | Poisoning rate | No defense | | ANP | | I-BAU | | FT+SAM | | NGF | | FE-tuning (Ours) | | FT-init (Ours) | | FST (Ours) | |
|---|---|---|---|---|---|---|---|---|---|---|---|---|---|---|---|---|---|
| | | C-Acc(↑) | ASR(↓) | C-Acc(↑) | ASR(↓) | C-Acc(↑) | ASR(↓) | C-Acc(↑) | ASR(↓) | C-Acc(↑) | ASR(↓) | C-Acc(↑) | ASR(↓) | C-Acc(↑) | ASR(↓) | C-Acc(↑) | ASR(↓) |
| BadNet | 5% | 94.09 | 99.99 | 91.62 | 0.52 | 91.72 | 1.84 | 91.71 | 2.99 | 90.60 | 2.58 | 91.54 | 3.70 | **93.24** | 20.02 | 93.17 | **0.00** |
| | 1% | 94.52 | 100 | 92.38 | 0.78 | 92.01 | 2.93 | 91.94 | 23.72 | 90.91 | 6.69 | 91.56 | 3.18 | **93.37** | 16.72 | 92.81 | **0.01** |
| | 0.5% | 94.79 | 100 | 84.81 | 12.12 | 91.14 | 10.01 | 92.58 | 43.04 | 92.37 | 7.41 | 92.37 | 7.41 | **93.9** | 79.32 | 93.63 | **0.02** |
| Blended | 5% | 94.91 | 99.76 | 91.68 | 5.31 | 90.9 | 43.51 | 92.24 | 40.80 | 91.42 | 21.24 | 91.08 | 11.19 | **93.28** | 60.94 | 92.87 | **3.07** |
| | 1% | 95.13 | 98.12 | 90.31 | 1.18 | 90.05 | 24.57 | 91.72 | 14.59 | 91.50 | 10.73 | 92.03 | 8.50 | **93.88** | 61.23 | 93.59 | **0.19** |
| | 0.5% | 94.45 | 92.46 | 90.93 | 7.92 | 90.91 | 15.73 | 91.50 | 29.32 | 91.14 | 10.03 | 91.84 | 6.48 | **93.71** | 49.80 | 93.15 | **0.06** |
| WaNet | 5% | 91.14 | 96.18 | 91.43 | 0.63 | 90.84 | 1.20 | 91.02 | 1.51 | 91.02 | 1.12 | 90.17 | 1.12 | 91.56 | 0.86 | 91.56 | **0.26** |
| | 1% | 90.71 | 70.13 | 91.02 | 0.64 | 91.63 | **0.39** | 91.17 | 1.10 | 89.88 | 1.07 | 89.59 | 1.20 | **92.06** | 0.67 | 91.83 | 0.51 |
| | 0.5% | 90.75 | 16.56 | 90.28 | 0.43 | 90.38 | 1.22 | 91.22 | 0.67 | 90.31 | 1.04 | 89.70 | 1.13 | **92.21** | 0.69 | 91.70 | 0.78 |
| SSBA | 5% | 94.54 | 97.39 | 91.64 | 1.17 | 90.68 | 6.76 | 92.64 | 2.69 | 91.05 | 2.57 | 91.49 | 11.21 | 93.38 | 43.67 | **93.48** | **0.27** |
| | 1% | 94.83 | 79.54 | 90.21 | 0.59 | 89.69 | 4.66 | 92.35 | 2.35 | 91.02 | 4.99 | 91.99 | 5.58 | **93.51** | 21.04 | 93.32 | **0.56** |
| | 0.5% | 94.50 | 50.20 | 90.00 | 0.87 | 90.19 | 1.62 | 92.15 | 1.98 | 91.11 | 2.29 | 91.31 | 2.50 | **93.41** | 6.69 | 92.97 | **0.04** |
| SIG | 5% | 94.62 | 99.42 | **94.36** | 1.89 | 90.99 | 84.97 | 92.05 | 1.96 | 91.18 | 25.53 | 91.59 | 21.66 | 93.61 | 55.76 | 93.24 | **0.02** |
| | 1% | 94.91 | 92.74 | 89.73 | 3.68 | 92.23 | 18.33 | 92.73 | 17.27 | 91.12 | 3.23 | 91.77 | 1.73 | **93.92** | 52.96 | 93.09 | **0.03** |
| | 0.5% | 94.69 | 93.76 | 89.77 | 1.44 | 90.29 | 37.80 | 92.56 | 61.93 | 91.22 | 17.10 | 91.53 | 3.01 | **93.61** | 69.08 | 93.18 | **0.01** |
| LC | 5% | 94.45 | 99.97 | 88.48 | 4.77 | 93.92 | 42.27 | 92.07 | 19.34 | 91.27 | 25.02 | 91.96 | 35.68 | **93.75** | 60.01 | 93.47 | **0.68** |
| | 1% | 94.33 | 99.16 | 87.97 | 4.18 | 93.01 | 10.33 | 92.00 | 27.88 | 91.57 | 59.08 | 91.7 | 64.97 | **93.54** | 89.86 | 93.51 | **0.30** |
| | 0.5% | 94.89 | 100 | 92.45 | 98.17 | 91.40 | 81.94 | 92.39 | 84.37 | 91.07 | 51.96 | 91.65 | 22.66 | **93.83** | 96.16 | 93.44 | **1.71** |
| Adaptive-Blend | 0.3% | 94.86 | 83.03 | 92.89 | 3.85 | 91.95 | 15.49 | 93.61 | 42.25 | 93.24 | 28.52 | 93.68 | 10.34 | **94.93** | 48.67 | 94.35 | **1.37** |
| Average | | 94.06 | 87.81 | 90.63 | 7.90 | 91.26 | 54.46 | 90.07 | 22.09 | 91.11 | 15.00 | 91.50 | 11.75 | **93.43** | 43.90 | 93.07 | **0.52** |
| Standard Deviation | | 1.44 | 21.59 | 2.07 | 22.07 | 1.06 | 143.15 | **0.61** | 22.69 | 0.67 | 16.95 | 0.92 | 15.74 | 0.70 | 30.43 | 0.70 | **0.78** |

Table 3: Defense results under various poisoning rates. The experiments are conducted on the GTSRB dataset with ResNet-18. All the metrics are measured in percentage (%). The best results are bold.

| Attack | Poisoning rate | No defense | | ANP | | I-BAU | | FT+SAM | | NGF | | FE-tuning (Ours) | | FT-init (Ours) | | FST (Ours) | |
|---|---|---|---|---|---|---|---|---|---|---|---|---|---|---|---|---|---|
| | | C-Acc(↑) | ASR(↓) | C-Acc(↑) | ASR(↓) | C-Acc(↑) | ASR(↓) | C-Acc(↑) | ASR(↓) | C-Acc(↑) | ASR(↓) | C-Acc(↑) | ASR(↓) | C-Acc(↑) | ASR(↓) | C-Acc(↑) | ASR(↓) |
| BadNet | 5% | 98.85 | 100 | **98.79** | 0.00 | 98.28 | 0.00 | 97.37 | 0.00 | 97.39 | 0.27 | 98.10 | 7.41 | 98.59 | 33.64 | 94.05 | 0.00 |
| | 1% | 98.56 | 100 | 98.58 | 0.00 | 97.71 | 0.18 | 99.05 | 0.68 | 98.18 | 0.00 | 96.1 | 0.02 | **98.80** | 99.88 | 96.31 | 0.00 |
| | 0.5% | 98.35 | 100 | **98.34** | 0.00 | 97.81 | 0.14 | 98.73 | 0.10 | 96.99 | 0.00 | 98.23 | 0.94 | 98.31 | 100 | 96.60 | 0.00 |
| Blended | 5% | 98.71 | 99.71 | 97.12 | **0.00** | 94.73 | 8.42 | **98.86** | 41.91 | 97.95 | 50.45 | 96.67 | 96.10 | 98.43 | 98.19 | 97.02 | 0.02 |
| | 1% | 96.60 | 97.12 | 91.12 | 40.97 | 97.43 | 73.43 | 97.04 | 76.35 | 96.97 | 61.46 | 97.83 | 88.66 | **98.56** | 93.25 | 97.43 | 3.24 |
| | 0.5% | 94.92 | 86.10 | 87.43 | 45.07 | 96.57 | 63.11 | 97.72 | 73.59 | 96.70 | 49.38 | 98.59 | 81.93 | **98.61** | 91.16 | 94.34 | 0.04 |
| WaNet | 5% | 96.20 | 96.89 | 97.55 | 0.02 | 93.86 | **0.00** | 87.41 | 0.70 | **98.50** | 1.70 | 84.17 | 0.68 | 91.49 | 3.04 | 95.00 | 0.02 |
| | 1% | 96.82 | 56.66 | 95.97 | **0.02** | 93.19 | 0.43 | 88.74 | 0.53 | **98.31** | 18.14 | 85.89 | 0.61 | 90.70 | 0.48 | 95.55 | 0.03 |
| | 0.5% | 97.46 | 28.86 | 97.74 | 0.03 | 94.62 | 0.02 | 88.62 | 0.14 | **97.87** | 0.17 | 81.28 | 0.31 | 93.44 | 2.15 | 97.17 | 0.00 |
| SSBA | 5% | 97.32 | 97.15 | 91.39 | 0.42 | 97.04 | 0.95 | 96.69 | 7.01 | 97.08 | 6.06 | 97.39 | 91.11 | **98.40** | 92.17 | 95.94 | 0.00 |
| | 1% | 96.75 | 92.27 | 93.60 | 0.06 | 92.60 | 1.88 | **98.07** | 62.05 | 96.76 | 14.06 | 97.62 | 77.68 | 97.96 | 82.78 | 96.39 | 0.12 |
| | 0.5% | 97.46 | 90.16 | 94.68 | 3.54 | 93.66 | 8.92 | 98.05 | 30.61 | 97.04 | 13.25 | 97.99 | 59.39 | **98.44** | 65.94 | 95.28 | 0.00 |
| SIG | 0.4% | 97.26 | 63.87 | 91.17 | 57.61 | 93.51 | 2.90 | 98.72 | 5.35 | 97.58 | 45.27 | 96.89 | 26.18 | **98.88** | 93.14 | 95.76 | 0.45 |
| | 0.25% | 98.56 | 64.23 | 96.18 | 29.41 | 97.50 | 5.37 | 97.41 | 2.82 | 96.72 | 29.16 | 97.79 | 68.89 | **98.39** | 84.07 | 96.31 | 0.87 |
| | 0.15% | 98.01 | 49.25 | 96.10 | 49.25 | 97.56 | 3.80 | 97.98 | 2.71 | 96.43 | 5.71 | 95.74 | 38.40 | **98.84** | 63.37 | 98.00 | 1.04 |
| Adaptive-Blend | 0.3% | 96.58 | 92.43 | 91.76 | 25.50 | 91.70 | **0.25** | 98.60 | 60.79 | **98.95** | 50.22 | 98.89 | 51.68 | 98.89 | 69.49 | 97.63 | 0.68 |
| Average | | 97.40 | 82.13 | 94.98 | 15.71 | 95.50 | 10.77 | 96.18 | 22.68 | **97.47** | 21.47 | 95.19 | 43.87 | 97.29 | 67.04 | 96.17 | 0.41 |
| Standard Deviation | | 1.05 | 21.57 | 3.36 | 20.94 | 2.14 | 22.06 | 3.88 | 28.77 | **0.72** | 21.66 | 5.59 | 36.72 | 2.66 | 36.47 | 1.11 | 0.81 |

Following the experimental setting in Section 3.2, we also conduct feature visualization of FST in Figure 3. It can be clearly observed that our FST significantly shifts backdoor features and makes them easily separable from clean features of the target class like FE-tuning. Combined with our outstanding performance, this further verifies that actively shifting learned backdoor features can effectively mitigate backdoor attacks. We also evaluate two initial methods, FE-tuning and FT-init. FE-tuning performs best on Tiny-ImageNet, leading to a little ASR drop by $0.18\%$ compared with FST. However, it also significantly sacrifices clean accuracy, leading to a large C-ACC drop by $8.1\%$ compared with FST. FST outperforms them by a large margin for backdoor robustness on CIFAR-10 and GTSRB. As we expected, FST achieves stable improvements in robustness and clean accuracy compared with FE-tuning and FT-init.

**Adaptive Attacks.** To further test the limit of FST, we conduct an adaptive attack evaluation for it. To bypass defense methods based on latent separation property, adaptive poisoning attacks [26] actively suppress the latent separation between backdoor and clean features by adopting an extremely low poisoning rate and adding regularization samples. Their evaluations also show that these attacks successfully bypass existing strong latent separation-based defenses. Hence, we believe it is also equally a powerful adaptive attack against our FST method. We evaluate our method against adaptive attacks. The results of the Adaptive-Blend attack are shown in Table 2, 3. The results of more adaptive attacks and various regularization samples are shown in *Appendix* C.4. We could observe that FST could still effectively purify such an adaptive attack achieving an average drop in ASR by $81.66\%$ and $91.75\%$ respectively for the CIFAR-10 and GTSRB.

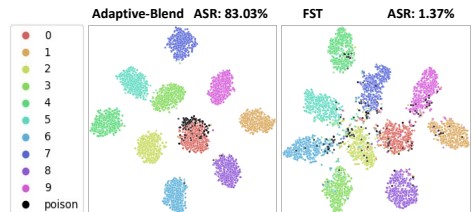

Figure 4: The T-SNE visualizations of Adaptive-Blend attack (150 payload and 150 regularization samples). Each color denotes each class, and **Black** points represent backdoored samples. The targeted class is **0 (Red)**. The left figure represents the original backdoored model and the right represents the model purified with FST.

Table 4: Defense results under various poisoning rate settings. The experiments are conducted on the Tiny-ImageNet dataset. All the metrics are measured in percentage (%). The best results are bold.

| Attack | Poisoning rate | No defense | | I-BAU | | FT+SAM | | NGF | | FE-tuning (Ours) | | FT-init (Ours) | | FST (Ours) | |
|---|---|---|---|---|---|---|---|---|---|---|---|---|---|---|---|
| | | C-Acc(↑) | ASR(↓) | C-Acc(↑) | ASR(↓) | C-Acc(↑) | ASR(↓) | C-Acc(↑) | ASR(↓) | C-Acc(↑) | ASR(↓) | C-Acc(↑) | ASR(↓) | C-Acc(↑) | ASR(↓) |
| BadNet | 5% | 85.17 | 100 | 76.33 | 80.45 | **81.17** | 81.93 | 78.29 | 55.39 | 71.19 | **0.00** | 80.20 | 15.21 | 79.23 | 1.41 |
| | 1% | 85.19 | 100 | 81.51 | 95.49 | **82.24** | 98.38 | 78.63 | 34.57 | 71.67 | **0.00** | 80.66 | 1.24 | 79.88 | 0.07 |
| | 0.5% | 85.42 | 99.97 | **81.03** | 84.19 | 80.06 | 60.50 | 79.04 | 27.02 | 72.16 | 0.00 | 80.79 | 0.03 | 79.82 | 0.00 |
| Blended | 5% | 85.30 | 99.88 | 76.82 | 86.74 | **81.88** | 91.37 | 78.85 | 86.99 | 71.96 | 0.00 | 80.58 | 0.00 | 79.78 | 0.00 |
| | 1% | 85.44 | 98.46 | **82.55** | 73.41 | 81.78 | 92.12 | 78.8 | 84.47 | 71.85 | 0.00 | 80.41 | 0.00 | 79.89 | 0.00 |
| | 0.5% | 85.49 | 95.49 | 79.19 | 71.37 | **80.96** | 76.10 | 78.85 | 76.57 | 71.78 | 0.00 | 80.61 | 0.00 | 80.02 | 0.00 |
| SSBA | 5% | 84.27 | 99.27 | **82.16** | 66.25 | 78.55 | **0.02** | 77.83 | 21.41 | 70.37 | 0.05 | 79.50 | 0.50 | 78.94 | 0.10 |
| | 1% | 85.11 | 89.05 | **82.60** | 68.51 | 82.14 | 21.51 | 78.56 | 13.24 | 71.46 | 0.01 | 80.35 | 0.07 | 79.73 | 0.06 |
| | 0.5% | 85.60 | 76.55 | 82.35 | 48.80 | **82.43** | 4.29 | 78.77 | 8.89 | 71.13 | 0.02 | 80.35 | 0.04 | 79.77 | 0.04 |
| Average | | 85.22 | 95.41 | 80.50 | 70.02 | **81.25** | 58.47 | 78.62 | 45.39 | 71.51 | **0.01** | 80.38 | 1.90 | 79.67 | 0.19 |
| Standard Deviation | | 0.39 | 7.93 | 2.47 | 13.66 | 1.26 | 39.36 | 0.37 | 31.08 | 0.55 | **0.02** | 0.38 | 5.01 | **0.35** | 0.46 |

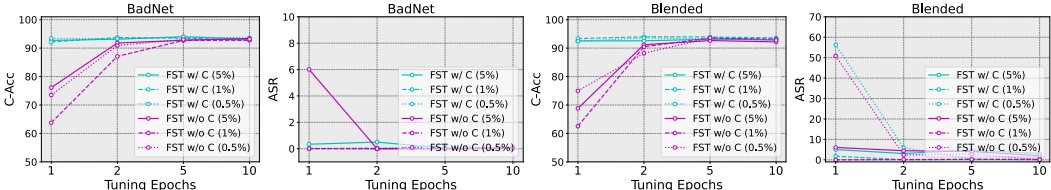

Figure 5: The experimental results with and without projection constraint (w/ C and w/o C, respectively). We demonstrate two types of backdoor attacks, namely the BadNet and Blended, with three different poisoning rates (5%, 1%, and 0.5%). The experiments are conducted with varying tuning epochs.

To explain why adaptive attacks fail, we provide TSNE visualizations of learned features from backdoored models. We show the results in Figure 4. We can first observe that adaptive attacks significantly reduce latent separability. Clean and backdoor features are tightly tangled. FST effectively shifts backdoor features and makes them easily separable from the clean features of the target class. Therefore, the feature extractor will no longer confuse backdoor features with clean features of the target class in the feature space. This leads to the subsequent simple linear classifier being difficult to be misled by backdoor samples, resulting in more robust classification.

## 5.3 Ablation Studies of FST

Below, we perform ablation studies on our proposed FST to analyze its efficiency and sensitivity to hyperparameters, tuning sizes, and architectures. This further demonstrates the effectiveness of FST in practical scenarios.

**Efficiency analysis.** we evaluate the backdoor defense performance of our method and 4 tuning strategies (FT+SAM, NGF, FE-tuning, and FT-init) across different tuning epochs. We test against 4 types of backdoor attacks including (BadNet, Blended, SSBA, and LC) with 1% poisoning rate and take experiments on CIFAR-10 and ResNet-18 models (see Figure 6). Notably, FST can effectively purify the backdoor models with much fewer epochs, reducing backdoor ASR below 5%. This demonstrates that our method also significantly improves tuning efficiency. We also find that added projection constraint also helps FST converge. We offer the performance of FST's entire tuning process on CIFAR-10 and ResNet-18, with or without the projection term. The results of the BadNet and Blended are shown in Figure 5. We could clearly observe that the projection stabilizes the tuning process of FST. The projection term helps FST quickly converge, particularly in terms of accuracy. We leave the analysis with more attacks in *Appendix* C.5.

**Sensitivity analysis on $\alpha$.** We evaluate the defense performance of FST with various $\alpha$ in Eq 1. We conduct experiments on CIFAR-10 and ResNet-18 and test against BadNet and Blended attacks with 5% and 1%. The results are shown in (a) and (b) of Figure 7. We can observe that as the $\alpha$ increases from 0 to 0.1, the backdoor ASR results rapidly drop below 5%. As we further increase the $\alpha$, our method maintains a stable defense performance (ASR < 4%) with only a slight accuracy degradation (< 1.5%). It indicates that FST is not sensitive to $\alpha$. The results further verify that FST could achieve a better trade-off between backdoor robustness and clean accuracy.

**Sensitivity analysis on tuning dataset size.** Here, we evaluate the FST under a rigorous scenario with limited access to clean tuning samples. We test our method on the CIFAR-10 using tuning datasets of varying sizes, ranging from 0.1% to 2%. Figure 7 shows that FST consistently performs

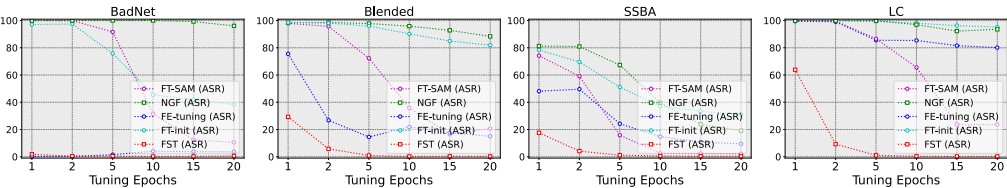

Figure 6: The ASR results of four backdoor attacks with varying tuning epochs of tuning methods.

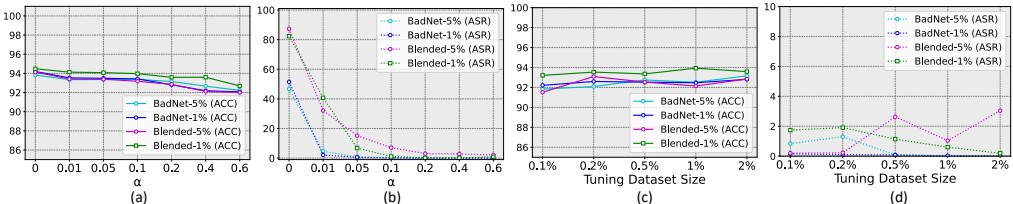

Figure 7: (a) and (b) show C-ACC and ASR of FST with various $\alpha$. (c) and (d) show the C-ACC and ASR of various sizes of tuning datasets. Experiments are conducted on CIFAR-10 dataset with ResNet-18.

well across various tuning data sizes. Even if the tuning dataset is reduced to only $0.1\%$ of the training dataset (50 samples in CIFAR-10), our FST can still achieve an overall ASR of less than $2\%$.

**Analysis on model architecture.** We extend the experiments on other model architectures including VGG19-BN [31], ResNet-50 [11], and DenseNet161 [12] which are widely used in previous studies [14, 37, 42]. Here we show the results of BadNet and Blended attacks with $5\%, 1\%$, and $0.5\%$ poisoning rates on the CIFAR-10 dataset. Notably, the structure of VGG19-BN is slightly different where its classifier contains more than one linear layer. Our initial experiments show that directly applying FST to the last layer fails to purify backdoors. Therefore, we simply extend our methods to the whole classifier and observe a significant promotion. We leave the remaining attacks and more implementation details in the *Appendix* D.2. The results presented in Figure 8 show that our FST significantly enhances backdoor robustness for all three architectures by reducing ASR to less than $5\%$ on average. This suggests that our approach is not influenced by variations in model architecture.

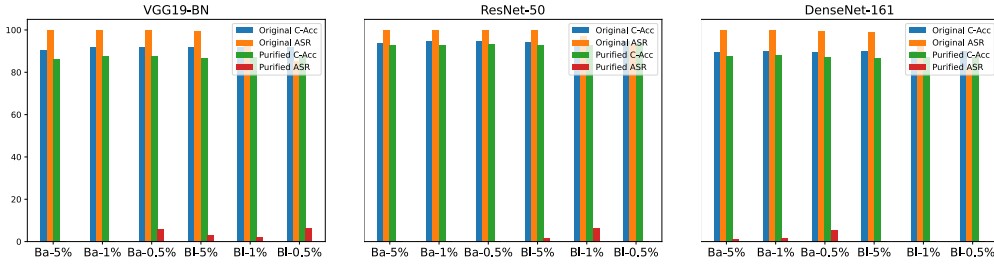

Figure 8: The purification performance against 3 different model architectures on the CIFAR-10 dataset, where Ba- is short for BadNet and Bl- is short for Blended.

## 6 Conclusion and limitations

In this work, we concentrate on practical Fine-tuning-based backdoor defense methods. We conduct a thorough assessment of widely used tuning methods, vanilla FT and LP. The experiments show that they both completely fail to defend against backdoor attacks with low poisoning rates. Our further experiments reveal that under low poisoning rate scenarios, the backdoor and clean features from the compromised target class are highly entangled together, and thus disentangling the learned features is required to improve backdoor robustness. To address this, we propose a novel defense approach called Feature Shift Tuning (FST), which actively promotes feature shifts. Through extensive evaluations, we demonstrate the effectiveness and stability of FST across various poisoning rates, surpassing existing strategies. However, our tuning methods assume that *the defender would hold a clean tuning set* which may not be feasible in certain scenarios. Additionally, *they also lead to a slight compromise on accuracy in large models though achieving robustness*. This requires us to pay attention to protect learned pretraining features from being compromised during robust tuning [7, 16, 19, 39].

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

# A  Social Impact

Deep Neural Networks (DNNs) are extensively applied in today's society especially for some safety-critical scenarios like autonomous driving and face verification. However, the data-hungry nature of these algorithms requires operators to collect massive amounts of data from diverse sources, making source tracing difficult and increasing the risk of potential malicious issues. For example, attackers can blend poisoned data into benign samples and embed backdoors into models without training control, posing a significant threat to model deployment. Therefore, to mitigate these risks, defenders must remove potential backdoors from models before real-world deployment, ensuring safety and trustworthiness. Our work focuses on a lightweight plug-and-play defense strategy applicable in real scenarios with minimal modifications to existing pipelines. We hope to appeal to the community to prioritize practical defensive strategies that enhance machine learning security.

# B  Experimental Settings

## B.1  Datasets and Models.

Following previous works [17, 36, 37, 38] in backdoor literature, we conduct our experiments on four widely used datasets including CIFAR-10, GTSRB, Tiny-ImageNet, and CIFAR-100.

- CIFAR-10 and GTSRB are two widely used datasets in backdoor literature containing images of $32 * 32$ resolution of 10 and 43 categories respectively. Following [37, 41], we separate $2\%$ clean samples from the whole training dataset for backdoor defense and leave the rest training images to implement backdoor models. For these two datasets, we utilize the ResNet-18 to construct the backdoor models.

- CIFAR-100 and Tiny-ImageNet are two datasets with larger scales compared to the CIFAR-10 and GTSRB which contain images with $64 * 64$ resolution of 100 and 200 categories respectively. For these two datasets, we enlarge the split ratio and utilize $5\%$ of the training dataset as backdoor defense since a smaller defense set is likely to hurt the model performance. For these two datasets, we utilize the pre-trained SwinTransformer (pre-trained weights on ImageNet are provided by *PyTorch*) to implement backdoor attacks since we find that training these datasets on ResNet-18 from scratch would yield a worse model performance with C-Acc ($< 70\%$) on average and therefore is not practical in real scenarios.

## B.2  Attack Configurations

We conducted all the experiments with 4 NVIDIA 3090 GPUs.

We implement 6 representative poisoning-based attacks and an adaptive attack called Adaptive-Blend [26]. For 6 representative attacks, most of them are built with the default configurations[2] in BackdoorBench [36]. For the BadNet, we utilize the checkerboard patch as backdoor triggers and stamp the pattern at the lower right corner of the image; for the Blended, we adopt the Hello-Kitty pattern as triggers and set the blend ratio as $0.2$ for both training and inference phase; for WaNet, we set the size of the backward warping field as 4 and the strength of the wrapping field as 0.5; for SIG, we set the amplitude and frequency of the sinusoidal signal as $40$ and $6$ respectively; for SSBA and LC, we adopt the pre-generated invisible trigger from BackdoorBench. For the extra adaptive attack, we utilize the official implementation [3] codes and set both the poisoning rate and cover rate as $0.003$ following the original paper. The visualization of the backdoored images is shown in Figure 9.

For CIFAR-10 and GTSRB, we train all the backdoor models with an initial learning rate of $0.1$ except for the WaNet since we find a large initial learning rate would make the attack collapse, and therefore we decrease the initial learning rate to $0.01$. All the backdoor models are trained for 100 epochs and 50 epochs for CIFAR-10 and GTSRB respectively. For CIFAR-100 and Tiny-ImageNet, we adopt a smaller learning rate of $0.001$ and fine-tune each model for 10 epochs since the SwinTransformer is already pre-trained on ImageNet and upscale the image size up to $224 * 224$ before feeding the image to the network.

---

[2]https://github.com/SCLBD/backdoorbench
[3]https://github.com/Unispac/Circumventing-Backdoor-Defenses

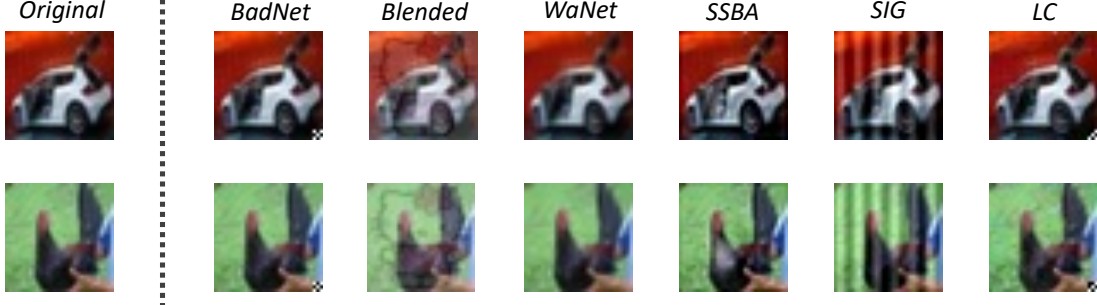

Figure 9: Example images of backdoored samples from CIFAR-10 dataset with 6 attacks.

### B.3    Baseline Defense Configurations

We evaluate 4 tuning-based defenses and 2 extra state-of-the-art defense strategies including both ANP and I-BAU for comparison. For tuning-based defenses, we mainly consider 2 recent works including FT+SAM and NGF, and we also compare another 2 baseline tuning strategies including FE-tuning and FT-init proposed in our paper. For all defense settings, we set the batch size as 128 on CIFAR10 and GTSRB and set the batch size as 32 on CIFAR-100 and Tint-ImageNet due to the memory limit.

- FT+SAM: Upon completion of our work, the authors of [42] had not yet made their source code publicly available. Therefore, we implemented a simplified version of their FT-SAM algorithm, where we replaced the optimizer with SAM in the original FT algorithm and called it FT+SAM. For both CIFAR-10 and GTSRB, we set the initial learning rate as 0.01 and fine-tune models with 100 epochs. We set the $\rho$ as 8 and 10 for CIFAR-10 and GTSRB respectively since we find the original settings ($\rho = 2$ for CIFAR-10 and $\rho = 8$ for GTSRB) are not sufficient for backdoor purification in our experiments. For CIFAR-100 and Tiny-ImageNet, we set the initial learning rate as 0.001 and $\rho$ as 6, and fine-tune the backdoor model for 20 epochs for fair comparison.

- NGF: We adopt the official implementation [4] for NGF. For CIFAR-10 and GTSRB, we set the tuning epochs as 100 and the initial learning rate as 0.015 and 0.05 respectively. While for CIFAR-100 and Tiny-ImageNet, we set the tuning epochs as 20 and the initial learning rate as 0.002.

- FE-tuning: For FE-tuning, we first re-initialize and freeze the parameters in the head. We then only fine-tune the remaining feature extractor. For CIFAR-10 and GTSRB, we set the initial learning rate as 0.01 and fine-tune the backdoor model with 100 epochs; while for CIFAR-100 and Tiny-ImageNet, we set the initial learning rate as 0.005 and fine-tune the backdoor model with 20 epochs.

- FT-init: For FT-init, we randomly re-initialize the linear head and fine-tune the whole model architecture. For CIFAR-10 and GTSRB, we set the initial learning rate as 0.01 and fine-tune the backdoor model with 100 epochs; while for CIFAR-100 and Tiny-ImageNet, we set the initial learning rate as 0.005 and fine-tune the backdoor model with 20 epochs.

- ANP: We follow the implementation in BackdoorBench and set the perturbation budget as 0.4 and the trade-off coefficient as 0.2 following the original configuration. We find that within a range of thresholds, the model performance and backdoor robustness are related to the selected threshold. Therefore, we set a threshold range (from 0.4 to 0.9) and present the purification results with low ASR and meanwhile maintain the model's performance.

- I-BAU: We follow the implementation in BackdoorBench and set the initial learning rate as $1e^{-4}$ and utilize 5 iterations for fixed-point approximation.

---

[4] https://github.com/kr-anonymous/ngf-animus

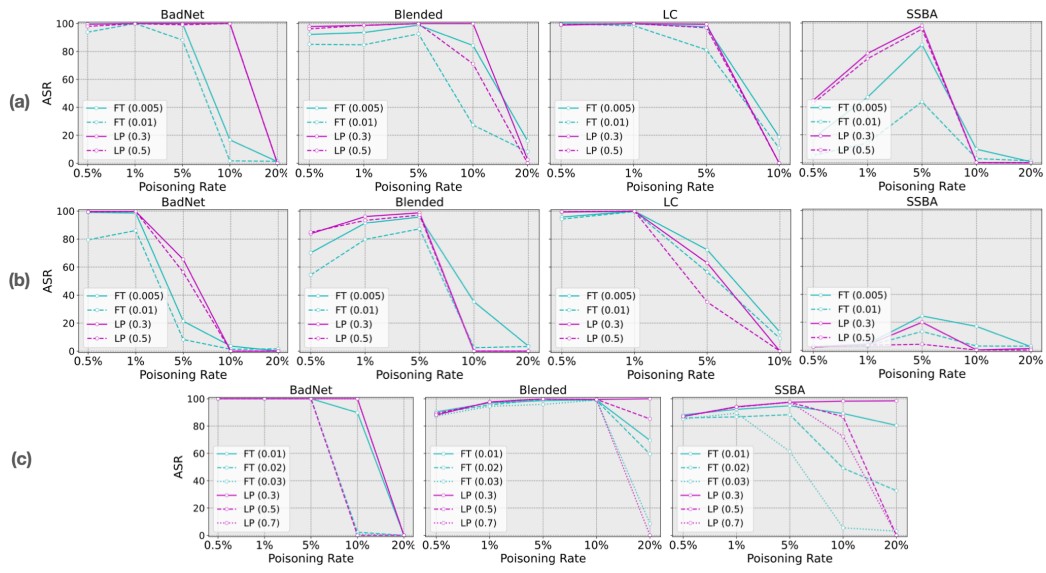

Figure 10: The Evaluation of vanilla FT and LP: (a) ResNet-50 on CIFAR-10. (b) Dense-161 on CIFAR-10. (c) ResNet-18 on GTSRB.

## C    Additional Experimental results

### C.1    Additional Results of Revisiting Fine-tuning

In this section, we provide additional experimental results for Section 3 to explore the potential influence of the dataset and model selection. Specifically, in addition to our initial experiments of revisiting fine-tuning on CIFAR-10 with ResNet-18, we further vary the model capacity (ResNet-50 on CIFAR-10), the model architecture (DenseNet-161 on CIFAR-10), and the dataset (ResNet-18 on GTSRB). As mentioned in Section 3.1, we mainly focus on defense performance with a satisfactory clean accuracy level (92% on CIFAR-10, 97% on GTSRB). We tune hyperparameters based on this condition. All the experimental results are shown in Figure 10 respectively. These additional results also demonstrate that Vanilla FT and LP could purify backdoored models for high poisoning rates but fail to defend against low poisoning rates attacks. The only exception is the SSBA results since the original backdoored models have a relatively low ASR, as mentioned in Section 3.1.

### C.2    Additional Results of High Poisoning Rates

Our previous experiments in Section 5.2 have demonstrated our FST's superior defense capacity against backdoor attacks with low poisoning rates. In this section, we further extend our attack scenarios with more poisoning samples by increasing the poisoning rate to 10%, 20%, and 30%. We conduct experiments on CIFAR-10 and GTSRB with ResNet-18 and present the experimental results in Table 5. We observe that the FST could easily eliminate the embedded backdoor as expected while preserving a high clean accuracy of the models.

### C.3    Additional Results on CIFAR-100 Dataset

We evaluate our FST on the CIFAR-100 dataset with the results shown in Table 6. Since some attacks show less effectiveness under a low poisoning rate with ASR < 25%, we hence only report the results where the backdoor attack is successfully implemented (original ASR ≥ 25%). We note that our FST could achieve excellent purification performance across all attack types on the CIFAR-100 dataset with an average ASR 0.36% which is 65.9% and 9.46% lower than the ASR of two other tuning strategies, FT+SAM and NGF respectively. Although the FE-tuning could achieve a lower

Table 5: Defense results under high poisoning rate settings. All the metrics are measured in percentage (%).

| Attack | Poisoning rate | CIFAR-10 | | | | GTSRB | | | |
|---|---|---|---|---|---|---|---|---|---|
| | | No defense | | FST | | No defense | | FST | |
| | | C-Acc(↑) | ASR(↓) | C-Acc(↑) | ASR(↓) | C-Acc(↑) | ASR(↓) | C-Acc(↑) | ASR(↓) |
| BadNet | 10% | 93.11 | 100 | 92.29 | 0.01 | 94.83 | 100 | 94.98 | 0.03 |
| | 20% | 92.80 | 100 | 91.82 | 0.30 | 97.81 | 100 | 94.09 | 0.01 |
| | 30% | 91.55 | 100 | 90.91 | 0.00 | 96.62 | 100 | 94.89 | 0.01 |
| Blended | 10% | 94.36 | 99.93 | 93.10 | 0.34 | 96.33 | 97.40 | 96.52 | 0.00 |
| | 20% | 94.21 | 100 | 92.97 | 0.23 | 91.96 | 98.56 | 95.53 | 0.02 |
| | 30% | 93.64 | 100 | 92.54 | 3.33 | 98.46 | 99.97 | 96.37 | 0.00 |
| WaNet | 10% | 90.86 | 97.26 | 92.63 | 0.14 | 97.08 | 94.21 | 95.61 | 0.02 |
| | 20% | 90.12 | 98.73 | 91.19 | 0.19 | 97.10 | 98.36 | 95.65 | 0.02 |
| | 30% | 80.58 | 97.32 | 90.37 | 0.71 | 94.20 | 99.63 | 93.42 | 0.03 |
| SSBA | 10% | 94.34 | 98.91 | 93.41 | 0.39 | 97.26 | 99.32 | 96.67 | 0.02 |
| | 20% | 93.47 | 99.66 | 92.72 | 0.23 | 96.42 | 96.15 | 96.03 | 0.01 |
| | 30% | 93.27 | 99.97 | 92.07 | 0.11 | 97.71 | 99.20 | 97.03 | 0.00 |

Table 6: Defense results under various poisoning rate settings. The experiments are conducted on the CIFAR-100 dataset. All the metrics are measured in percentage (%). The best results are bold.

| Attack | Poisoning rate | No defense | | I-BAU | | FT+SAM | | NGF | | FE-tuning (Ours) | | FT-init (Ours) | | FST (Ours) | |
|---|---|---|---|---|---|---|---|---|---|---|---|---|---|---|---|
| | | C-Acc(↑) | ASR(↓) | C-Acc(↑) | ASR(↓) | C-Acc(↑) | ASR(↓) | C-Acc(↑) | ASR(↓) | C-Acc(↑) | ASR(↓) | C-Acc(↑) | ASR(↓) | C-Acc(↑) | ASR(↓) |
| BadNet | 5% | 85.47 | 100 | **83.10** | 99.89 | 82.89 | 99.41 | 70.22 | 0.68 | 72.19 | 0.05 | 80.02 | 0.03 | 78.99 | **0.00** |
| | 1% | 85.85 | 99.96 | **83.27** | 99.22 | 83.00 | 95.30 | 70.11 | 0.49 | 72.25 | 0.03 | 80.33 | 0.03 | 79.54 | **0.00** |
| | 0.5% | 84.71 | 99.61 | 82.70 | 92.78 | **83.02** | 87.89 | 69.95 | 0.47 | 71.75 | **0.00** | 80.63 | 0.02 | 80.11 | 0.01 |
| Blended | 5% | 85.75 | 100 | 83.11 | 99.99 | **83.14** | 97.86 | 70.20 | 20.89 | 72.25 | **0.54** | 80.27 | 0.87 | 80.36 | 0.83 |
| | 1% | 85.73 | 99.93 | 83.14 | 99.48 | 83.14 | 93.70 | 69.85 | 14.72 | 72.52 | **0.53** | 80.55 | 0.82 | 80.57 | 0.74 |
| | 0.5% | 85.71 | 99.71 | **83.12** | 98.81 | 82.79 | 95.97 | 69.95 | 24.58 | 72.62 | 0.68 | 80.58 | 0.45 | 80.2 | **0.31** |
| SSBA | 5% | 85.13 | 92.79 | 82.94 | 29.49 | **83.06** | 4.12 | 69.18 | 0.36 | 71.94 | 0.20 | 80.40 | 0.23 | 79.97 | **0.17** |
| | 1% | 85.15 | 54.52 | **83.55** | 4.16 | 83.01 | 0.19 | 70.64 | 0.32 | 72.05 | 0.19 | 79.33 | 0.20 | 80.4 | 0.20 |
| SIG | 1% | 85.48 | 40.12 | **82.98** | 29.00 | 82.63 | 21.71 | 69.88 | 25.68 | 72.79 | **0.71** | 80.14 | 0.77 | 79.91 | 0.83 |
| Average | | 85.44 | 87.40 | **83.10** | 72.54 | 82.96 | 66.24 | 70.00 | 9.80 | 72.26 | 32.56 | 80.25 | 0.38 | 80.01 | **0.34** |
| Standard Deviation | | 0.38 | 23.13 | 0.23 | 39.47 | **0.17** | 43.67 | 0.39 | 11.48 | 0.33 | **0.29** | 0.40 | 0.36 | 0.49 | 0.36 |

ASR compared to FST, we note that its C-Acc gets hurt severely since it freezes the re-initialized linear head during fine-tuning which restricts its feature representation space. For the other two state-of-the-art defenses, we find that they are less effective in purifying the larger backdoor models.

We further observe that the FT-init could achieve comparable purification results as the FST with even a slightly higher C-Acc. Compared to our previous experiments on the small-scale dataset (CIFAR-10 and GTSRB) and model (ResNet-18), we find that FT-init is more effective on the large model (SwinTransformer) with the large-scale dataset (CIFAR-100 and Tiny-ImageNet) which decreases the average ASR by 32.31%.

## C.4 Additional Results of Adaptive Attacks

In addition to the Adaptive-Blend attack, we also provide evaluations of a parallel attack proposed in [26] called Adaptive-Patch. To further reduce latent separability and improve adaptiveness against latent separation-based defenses, we also use more regularization samples, following ablation study of Section 6.3 [26]. The experimental results are presented in Table 7 and demonstrate that our FST could purify both attack types with various regularization samples. We also demonstrate a T-SNE visualization of the Adaptive-Patch in Figure 11. It aligns with the results of Adaptive-Blend attack.

To further assess stability of FST, we also test FST against training-control adaptive attacks [30]. The authors [30] utilize an adversarial network regularization during the training process to minimize differences

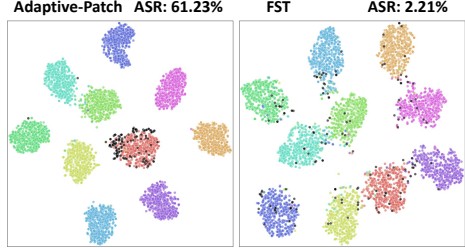

Figure 11: The T-SNE visualizations of Adaptive-Patch attack (150 payload and 300 regularization samples). Each color denotes each class, and **Black** points represent backdoored samples. The targeted class is **0 (Red)**. The left figure represents the original backdoored model and the right represents the model purified with FST.

between backdoor and clean features in latent representations. Since the authors do not provide source code, we follow their original methodology and implement their Adversarial Embedding attack with two types of trigger, namely the checkboard patch (Bypass-Patch) and Hello-Kitty pattern

Table 7: Defense results of Adaptive-Blend and Adaptive-Patch attacks with various regularization samples. The metrics C-ACC and ASR are measured in percentage.

| Attack | Regularization samples | No defense | | FST | |
|---|---|---|---|---|---|
| | | C-Acc(↑) | ASR(↓) | C-Acc(↑) | ASR(↓) |
| Adaptive-Patch | 150 | 94.55 | 96.77 | 93.58 | 0.28 |
| | 300 | 94.59 | 61.23 | 91.99 | 2.21 |
| | 450 | 94.52 | 54.23 | 91.41 | 5.42 |
| Adaptive-Blend | 150 | 94.86 | 83.03 | 94.35 | 1.37 |
| | 200 | 94.33 | 78.40 | 92.08 | 0.78 |
| | 300 | 94.12 | 68.99 | 92.29 | 1.39 |

Table 8: Defense results of Bypass attacks with three different poisoning rates.

| Attack | Poisoning rate | No defense | | FST | |
|---|---|---|---|---|---|
| | | C-Acc(↑) | ASR(↓) | C-Acc(↑) | ASR(↓) |
| Bypass-Patch | 5% | 89.81 | 96.28 | 87.85 | 0.02 |
| | 1% | 90.04 | 93.90 | 87.83 | 0.03 |
| | 0.5% | 89.50 | 58.83 | 87.61 | 0.73 |
| Bypass-Blend | 5% | 87.79 | 99.54 | 89.14 | 0.13 |
| | 1% | 89.70 | 83.66 | 88.11 | 0.08 |
| | 0.5% | 89.47 | 85.52 | 87.13 | 0.12 |

(Bypass-Blend). All the experimental results along with three poisoning rates are shown in Table 8. The results reveal that our FST could still mitigate the Bypass attack which emphasizes the importance of feature shift in backdoor purification.

### C.5 Additional Results of Projection Constraint Analysis

In this section, we first provide additional analysis of the projection constraint with more attacks (WaNet, SSBA, SIG, and LC) on the CIFAR-10 dataset. We show the experimental results in Figure 12. We get the same observations shown in Section 5.3, where the inclusion of the projection term plays a crucial role in stabilizing and accelerating the convergence process of the FST. This results in a rapid and satisfactory purification of the models within a few epochs.

## D Extra Ablation Studies

### D.1 Efficiency Analysis

We compare the backdoor purification efficiency of our FST with other tuning methods on the remaining three datasets including GTSRB, CIFAR-100, and Tiny-ImageNet. We select three

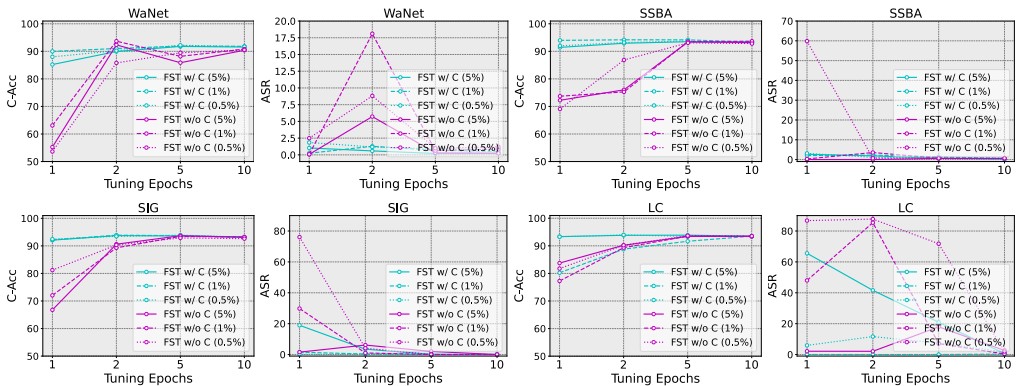

Figure 12: We demonstrate the experimental results with and without projection constraint (w/ C and w/o C, respectively) of four backdoor attacks, namely the WaNet, SSBA, SIG, and LC. The experiments are conducted with three poisoning rates (5%, 1%, and 0.5%) and varying tuning epochs.

representative attacks (BadNet, Blend, and SSBA) with poisoning rate $1\%$ which could be successfully implemented across three datasets and we present our experimental results in Figure 13, 14 and 15. The experimental results demonstrate that *our FST is efficient compared to the other* 4 *tuning-based backdoor defense which could constantly depress the ASR under a low-value range (usually* $< 5\%$*) with only a few epochs.* Besides, we also note that in the GTSRB dataset, both the ASR of FE-tuning and FT-init would increase as the tuning epoch increases indicating the model is gradually recovering the previous backdoor features. Our FST, however, maintains a low ASR along the tuning process which verifies the stability of our method.

## D.2 Diverse Model Architecture

We conduct comprehensive evaluations on three model architectures (VGG19-BN, ResNet-50, and DenseNet-161) on the CIFAR-10 dataset with all 6 representative poisoning-based backdoor attacks and one adaptive attack, and our experimental results are shown in Table 9, 10 and 11. During our initial experiments, we note that our method is less effective for VGG19-BN. One possible reason is that the classifier of VGG19-BN contains more than one layer which is slightly different from our previously used structure ResNet-18. Therefore, one direct idea is to extend our original last-layer regularization to all the last linear layers of VGG19-BN. For implementation, we simply change the original $\alpha \langle \boldsymbol{w}, \boldsymbol{w}^{ori} \rangle$ to $\alpha \sum_i \langle \boldsymbol{w_i}, \boldsymbol{w_i^{ori}} \rangle$ where $i$ indicates each linear layer. Based on this, we obtain an obvious promotion of backdoor defense performances (shown in Figure 16) without sacrificing clean accuracy.

Following the results in Table 9, our FST could achieve better and much more stable performance across all attack settings with an average ASR of $6.18\%$ and a standard deviation of $11.64\%$. Compared with the four tuning-based defenses, our FST could achieve $29\%$ lower on ASR average across all the attack settings; compared with the other two state-of-the-art defensive strategies, our FST achieves a much lower ASR while getting a much smaller C-Acc drop ($< 3.5\%$). For the other two architectures, we note that our FST could achieve the best performance across all attack settings with an average ASR of $2.7\%$ and maintain clean accuracy (the drop of C-Acc $< 1.9\%$).

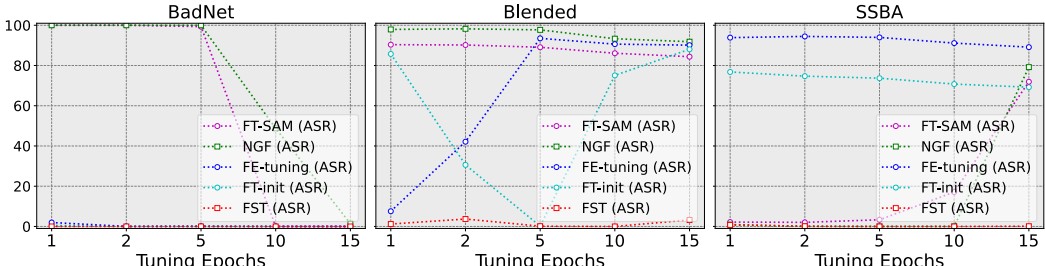

Figure 13: The ASR results of three representative attacks with various tuning epochs. Our experiments are conducted on GTSRB with ResNet-18.

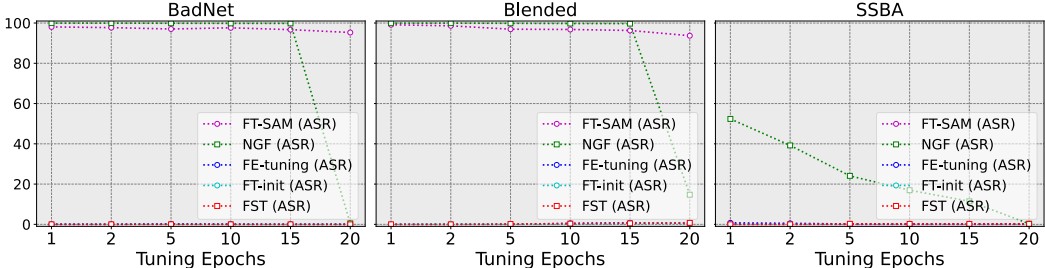

Figure 14: The ASR results of three representative attacks with various tuning epochs. Our experiments are conducted on CIFAR-100 with SwinTransformer.

Table 9: Defense results under various poisoning rates. The experiments are conducted on the CIFAR-10 dataset with VGG19-BN. All the metrics are measured in percentage (%). The best results are bold.

| Attack | Poisoning rate | No defense | | ANP | | I-BAU | | FT+SAM | | NGF | | FE-tuning (Ours) | | FT-init (Ours) | | FST (Ours) | |
|---|---|---|---|---|---|---|---|---|---|---|---|---|---|---|---|---|---|
| | | C-Acc(↑) | ASR(↓) | C-Acc(↑) | ASR(↓) | C-Acc(↑) | ASR(↓) | C-Acc(↑) | ASR(↓) | C-Acc(↑) | ASR(↓) | C-Acc(↑) | ASR(↓) | C-Acc(↑) | ASR(↓) | C-Acc(↑) | ASR(↓) |
| BadNet | 5% | 90.69 | 100 | 82.00 | 0.00 | 86.03 | 4.92 | 84.81 | 6.89 | 83.40 | 5.39 | 84.66 | 62.21 | 84.05 | 3.67 | **86.33** | **0.01** |
| | 1% | 91.86 | 100 | 85.42 | 4.57 | 86.47 | 99.71 | 84.89 | 76.71 | 81.78 | 8.81 | 85.60 | 95.71 | 84.09 | 77.99 | **87.30** | **0.00** |
| | 0.5% | 91.88 | 100 | 86.81 | 99.96 | 83.44 | 98.48 | 85.55 | 71.37 | 83.44 | 44.62 | 85.26 | 99.91 | 84.41 | 98.19 | **87.27** | **0.70** |
| Blended | 5% | 91.79 | 99.41 | 85.57 | 53.53 | **86.28** | 34.54 | 84.09 | 12.21 | 82.58 | 7.58 | 85.48 | 38.67 | 84.47 | 17.01 | 86.15 | **2.71** |
| | 1% | 92.07 | 93.87 | 86.67 | 39.30 | 84.10 | 18.32 | 85.61 | 31.12 | 84.42 | 16.12 | 85.43 | 29.44 | 84.77 | 22.41 | **86.83** | **2.31** |
| | 0.5% | 92.04 | 85.09 | **91.24** | 76.22 | 85.30 | 30.60 | 85.23 | 34.98 | 84.28 | 17.50 | 85.09 | 33.92 | 84.79 | 40.24 | 87.89 | **6.18** |
| WaNet | 5% | 88.11 | 94.00 | 89.38 | **0.69** | 81.52 | 1.29 | 88.75 | 1.36 | 88.77 | 1.78 | 89.29 | 8.22 | 89.20 | 10.31 | **89.66** | 1.69 |
| SSBA | 5% | 91.10 | 89.08 | **88.50** | **0.98** | 82.55 | 7.08 | 85.12 | 2.7 | 83.59 | 3.38 | 84.97 | 6.21 | 83.49 | 2.48 | 85.57 | 2.77 |
| | 1% | 91.85 | 40.60 | 90.01 | **1.66** | 84.65 | 3.13 | 85.36 | 2.58 | 83.62 | 2.42 | 84.91 | 2.94 | 85.00 | 2.90 | 87.71 | 1.79 |
| SIG | 5% | 91.74 | 97.41 | 89.85 | 2.58 | 82.70 | 2.51 | 85.07 | 6.10 | 83.20 | 0.62 | 85.19 | 23.98 | 84.74 | 1.90 | 86.06 | **0.01** |
| | 1% | 91.76 | 93.33 | **88.12** | 26.23 | 82.39 | 49.12 | 85.09 | 44.96 | 82.96 | 46.20 | 85.59 | 73.24 | 84.81 | 47.26 | 86.3 | 18.28 |
| | 0.5% | 92.00 | 82.42 | **89.33** | 16.96 | 81.56 | 40.89 | 85.03 | 9.29 | 83.95 | 20.47 | 85.81 | 32.22 | 85.51 | 33.41 | 86.84 | **2.88** |
| LC | 5% | 91.59 | 100 | 84.66 | 40.57 | **88.35** | 70.99 | 85.49 | 13.69 | 83.95 | 20.32 | 85.49 | 91.03 | 84.84 | 50.87 | 86.22 | **0.10** |
| | 1% | 91.79 | 100 | 84.16 | 97.26 | **87.72** | 99.76 | 83.97 | 36.27 | 83.77 | 51.67 | 85.25 | 99.68 | 84.81 | 97.09 | 86.70 | **0.88** |
| | 0.5% | 92.07 | 100 | 84.7 | 89.68 | 83.72 | 86.64 | 85.34 | 49.54 | 83.92 | 69.26 | 85.30 | 99.01 | 85.05 | 93.41 | **87.58** | 13.09 |
| Adaptive-Blend | 0.3% | 92.11 | 66.84 | 89.79 | 54.02 | 88.10 | 27.38 | 89.36 | **14.57** | 89.03 | 33.48 | 90.26 | 44.86 | **90.71** | 51.82 | 90.06 | 45.44 |
| Average | | 91.53 | 90.13 | **87.26** | 37.76 | 84.68 | 42.1 | 85.55 | 25.90 | 84.17 | 21.85 | 85.85 | 52.58 | 85.30 | 40.69 | 87.15 | **6.18** |
| Standard Deviation | | 0.99 | 16.01 | 2.65 | 36.94 | 2.29 | 37.44 | 1.45 | 24.45 | 1.96 | 21.07 | 1.57 | 35.98 | 1.90 | 35.18 | **1.24** | **11.64** |

Table 10: Defense results under various poisoning rates. The experiments are conducted on the CIFAR-10 dataset with ResNet-50. All the metrics are measured in percentage (%). The best results are bold.

| Attack | Poisoning rate | No defense | | ANP | | I-BAU | | FT+SAM | | NGF | | FE-tuning (Ours) | | FT-init (Ours) | | FST (Ours) | |
|---|---|---|---|---|---|---|---|---|---|---|---|---|---|---|---|---|---|
| | | C-Acc(↑) | ASR(↓) | C-Acc(↑) | ASR(↓) | C-Acc(↑) | ASR(↓) | C-Acc(↑) | ASR(↓) | C-Acc(↑) | ASR(↓) | C-Acc(↑) | ASR(↓) | C-Acc(↑) | ASR(↓) | C-Acc(↑) | ASR(↓) |
| BadNet | 5% | 94.02 | 100 | 88.42 | 0.56 | 91.65 | 2.33 | 90.29 | 4.17 | 87.87 | 3.14 | 91.63 | 2.14 | 92.84 | 3.38 | **93.02** | **0.24** |
| | 1% | 94.30 | 99.99 | 90.97 | 1.76 | 89.17 | 2.51 | 90.41 | 5.93 | 87.47 | 5.62 | 90.94 | 1.47 | 92.33 | 6.32 | **92.65** | **0.26** |
| | 0.5% | 94.75 | 99.86 | 87.16 | 6.89 | 90.42 | 2.03 | 90.95 | 3.96 | 87.73 | 4.18 | 91.28 | 1.99 | **93.22** | 4.44 | 93.03 | **0.87** |
| Blended | 5% | 94.38 | 99.62 | 91.95 | 2.22 | 91.69 | 6.72 | 90.35 | 14.53 | 87.29 | 7.58 | 91.28 | 10.23 | 92.79 | 52.24 | **92.82** | **2.20** |
| | 1% | 94.90 | 97.51 | 88.59 | 9.82 | 91.06 | 37.86 | 91.16 | 27.14 | 88.11 | 4.90 | 91.47 | 25.71 | **93.21** | 57.89 | 93.11 | **6.29** |
| | 0.5% | 94.08 | 90.54 | 89.95 | 20.79 | 90.99 | 42.61 | 90.37 | 17.38 | 87.98 | 10.28 | 91.39 | 25.91 | 92.57 | 52.23 | **93.71** | **0.3** |
| WaNet | 5% | 92.03 | 87.86 | 84.00 | 1.82 | 89.21 | 2.58 | 91.84 | 0.92 | 90.24 | 1.80 | 92.23 | 1.11 | **92.66** | **0.80** | 92.43 | 0.31 |
| | 1% | 91.11 | 73.81 | 91.79 | **0.79** | 88.36 | 1.68 | 91.09 | 0.89 | 90.28 | 1.40 | 92.02 | 0.89 | 92.21 | **0.67** | 92.21 | 0.39 |
| | 0.5% | 89.79 | 59.16 | 85.51 | 1.08 | 87.76 | **0.56** | 91.39 | 1.09 | 89.56 | 1.49 | 91.81 | 1.13 | **92.57** | 0.99 | 92.41 | 0.70 |
| SSBA | 5% | 93.81 | 97.57 | 88.01 | **0.36** | 91.27 | 6.49 | 90.29 | 1.56 | 87.93 | 2.38 | 90.93 | 8.76 | 92.19 | 13.20 | **92.23** | 0.48 |
| | 1% | 94.23 | 73.68 | 90.59 | 1.11 | 91.28 | 2.18 | 90.38 | 2.00 | 87.69 | 1.93 | 90.90 | 2.36 | **92.56** | 3.08 | 92.41 | **0.37** |
| | 0.5% | 94.34 | 43.42 | 90.19 | **0.72** | 91.78 | 1.20 | 91.06 | 1.86 | 87.66 | 2.27 | 91.06 | 1.86 | 92.74 | 2.82 | **92.96** | 1.17 |
| SIG | 5% | 94.49 | 98.82 | **92.97** | 7.83 | 91.89 | 6.76 | 90.90 | 4.88 | 87.42 | 0.51 | 91.06 | 30.16 | 92.94 | 65.34 | 92.43 | **0.02** |
| | 1% | 94.04 | 92.76 | **93.34** | 79.70 | 90.12 | 28.44 | 89.74 | 59.39 | 87.84 | 0.80 | 90.34 | 68.79 | 92.53 | 90.14 | 89.03 | **3.56** |
| | 0.5% | 94.68 | 86.91 | 92.93 | 78.67 | 91.64 | 7.79 | 90.89 | 11.67 | 87.36 | **0.24** | 90.23 | 19.88 | **92.94** | 53.93 | 89.17 | 9.82 |
| LC | 5% | 94.61 | 99.92 | **94.12** | 6.20 | 91.70 | 16.94 | 89.75 | **2.32** | 87.91 | 6.71 | 91.53 | 19.90 | 93.27 | 69.27 | 93.17 | 3.81 |
| | 1% | 94.30 | 99.87 | 89.17 | 15.49 | 91.50 | 17.57 | 90.60 | 11.71 | 87.90 | 11.84 | 91.35 | 18.48 | 92.83 | 89.16 | 92.6 | **1.51** |
| | 0.5% | 94.63 | 99.99 | 90.30 | 62.14 | 90.73 | **5.66** | 90.57 | 55.99 | 87.30 | 13.24 | 91.45 | 19.52 | **93.30** | 77.02 | 92.77 | 17.22 |
| Adaptive-Blend | 0.3% | 94.36 | 74.57 | 89.75 | 14.57 | 90.51 | 32.12 | 92.40 | 23.71 | 89.18 | 17.54 | 91.14 | 11.41 | **94.18** | 19.49 | 92.95 | **1.42** |
| Average | | 93.83 | 88.20 | 89.98 | 16.45 | 90.67 | 11.79 | 90.76 | 13.21 | 88.14 | 5.15 | 91.27 | 14.30 | **92.84** | 34.86 | 92.37 | **2.68** |
| Standard Deviation | | 1.35 | 16.20 | 2.66 | 26.25 | 1.22 | 13.54 | 0.66 | 17.55 | 0.94 | 4.91 | 0.50 | 16.56 | **0.47** | 33.62 | 1.21 | 4.32 |

Table 11: Defense results under various poisoning rates. The experiments are conducted on the CIFAR-10 dataset with DenseNet-161. All the metrics are measured in percentage (%). The best results are bold.

| Attack | Poisoning rate | No defense | | ANP | | I-BAU | | FT+SAM | | NGF | | FE-tuning (Ours) | | FT-init (Ours) | | FST (Ours) | |
|---|---|---|---|---|---|---|---|---|---|---|---|---|---|---|---|---|---|
| | | C-Acc(↑) | ASR(↓) | C-Acc(↑) | ASR(↓) | C-Acc(↑) | ASR(↓) | C-Acc(↑) | ASR(↓) | C-Acc(↑) | ASR(↓) | C-Acc(↑) | ASR(↓) | C-Acc(↑) | ASR(↓) | C-Acc(↑) | ASR(↓) |
| BadNet | 5% | 89.38 | 99.99 | 88.46 | 99.70 | 84.72 | 14.53 | 83.94 | 2.49 | 82.58 | 3.14 | 86.03 | 2.48 | 87.96 | 5.32 | 87.62 | **1.4** |
| | 1% | 89.86 | 99.93 | 85.34 | 97.42 | 84.45 | 28.07 | 83.09 | 3.76 | 84.53 | 11.73 | 85.74 | 1.97 | **88.37** | 43.50 | 88.22 | **1.63** |
| | 0.5% | 89.62 | 99.58 | **88.50** | 98.00 | 84.08 | 50.31 | 82.85 | **4.30** | 83.68 | 19.46 | 85.16 | 9.28 | 87.78 | 45.86 | 87.16 | 5.53 |
| Blended | 5% | 89.93 | 99.13 | 85.35 | 6.57 | 83.38 | 6.31 | 83.07 | 4.43 | 83.56 | 2.47 | 84.88 | 2.03 | 87.76 | 40.61 | 86.81 | **0.02** |
| | 1% | 89.82 | 92.69 | 83.62 | 64.44 | 86.51 | 4.97 | 82.50 | 3.76 | 83.82 | 2.84 | 84.74 | 1.62 | **87.85** | 23.34 | 87.19 | **0.47** |
| | 0.5% | 90.15 | 82.44 | 85.61 | 9.28 | 84.99 | 23.76 | 83.79 | 4.43 | 84.5 | 1.14 | 86.20 | 1.33 | **88.53** | 21.78 | 88.33 | **0.31** |
| WaNet | 5% | 82.76 | 64.86 | 83.77 | 1.79 | 81.96 | 3.48 | 80.51 | 1.71 | 84.03 | 1.51 | 83.68 | 2.51 | **85.22** | 1.70 | 85.02 | **0.92** |
| SSBA | 5% | 88.71 | 81.09 | 87.49 | 1.48 | 84.12 | 5.44 | 81.92 | 2.32 | 82.60 | 2.67 | 83.50 | 1.66 | 86.73 | 3.42 | 86.39 | **1.14** |
| SIG | 5% | 89.74 | 97.71 | 89.01 | 0.54 | 83.71 | 24.43 | 82.96 | 0.79 | 83.95 | 0.91 | 85.15 | 0.34 | 87.33 | 15.06 | 87.75 | **0.10** |
| | 1% | 89.04 | 95.34 | 86.9 | 7.91 | 83.33 | 23.77 | 82.78 | 7.76 | 83.85 | 4.92 | 85.06 | 4.14 | **87.01** | 27.91 | 86.16 | **4.11** |
| | 0.5% | 89.46 | 76.17 | 82.32 | 61.93 | 85.96 | 67.30 | 82.78 | 8.86 | 83.94 | 11.08 | 84.93 | 10.91 | **87.82** | 60.51 | 84.60 | **4.00** |
| LC | 5% | 89.56 | 99.71 | 89.39 | 98.16 | 87.19 | **1.68** | 82.19 | 4.21 | 83.62 | 4.81 | 84.53 | 5.37 | 87.47 | 3.96 | 86.46 | 3.67 |
| | 1% | 89.76 | 99.96 | 89.67 | 99.72 | 86.36 | 90.44 | 82.84 | 3.79 | 83.21 | 14.02 | 84.83 | 12.87 | 86.96 | 97.90 | 87.15 | **2.52** |
| | 0.5% | 88.24 | 99.04 | 88.40 | 98.52 | 81.40 | 60.78 | 79.64 | **5.83** | 81.72 | 21.26 | 82.38 | 19.17 | 85.64 | 74.03 | 84.63 | 13.68 |
| Adaptive-Blend | 0.3% | 88.18 | 51.77 | 85.74 | 23.56 | 86.22 | 7.87 | 86.89 | 15.19 | 83.70 | 2.37 | 85.41 | 4.05 | **89.10** | 3.95 | 86.41 | **1.33** |
| Average | | 88.95 | 89.29 | 86.64 | 51.27 | 84.56 | 26.91 | 82.69 | 5.71 | 83.69 | 8.94 | 84.62 | 5.26 | **87.44** | 35.26 | 86.66 | **2.72** |
| Standard Deviation | | 1.81 | 15.02 | 2.30 | 44.43 | 1.68 | 27.75 | 1.58 | 4.24 | **0.83** | 9.60 | 1.04 | 5.41 | 1.03 | 32.48 | 1.18 | **3.47** |

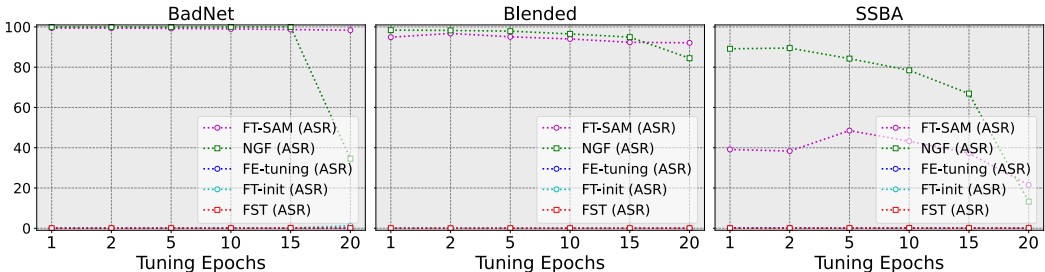

Figure 15: The ASR results of three representative attacks with various tuning epochs. Our experiments are conducted on Tiny-ImageNet with SwinTransformer.

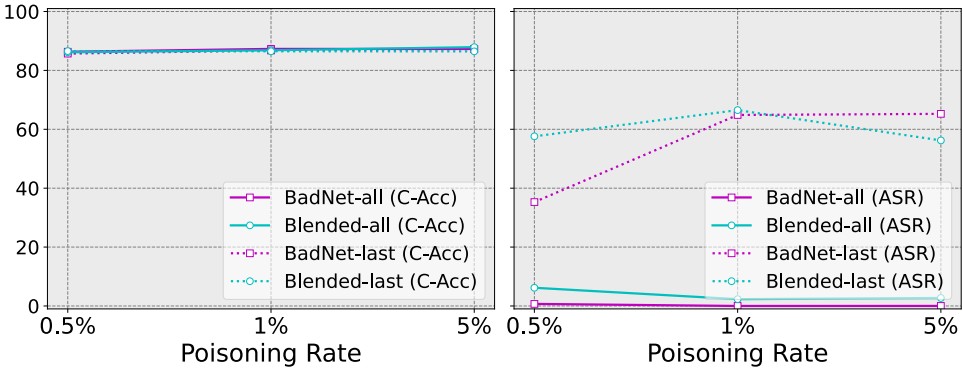

Figure 16: We compare regularizing the whole linear layers (denoted as -all) with regularizing only the last linear layer (denoted as -last). We evaluate on CIFAR-10 dataset using BadNet and Blended attacks with 3 poisoning rate settings. Experimental results demonstrate that we could achieve a superior purification performance by regularizing the whole linear layers than the last-layer-only regularization without sacrificing the model performance.

