# OpenReview forum: "Towards Stable Backdoor Purification through Feature Shift Tuning"
_NeurIPS.cc/2023/Conference — NeurIPS 2023 poster_

### Official Review · Reviewer_JBzP · 2023-07-05

**Soundness:** 4 excellent
**Presentation:** 3 good
**Contribution:** 3 good
**Rating:** 7
**Confidence:** 4

**Summary:**

Based on the observations that the previous FT and FP methods fail at low poisoning rate scenarios, this paper finds out the potential reasons in terms of clean and backdoor feature separation degree. It proposes FST to solve the problem by disentangling the features and validates its effectiveness with multiple backdoor attacks.

**Strengths:**

1. The paper is well-written and easy to read. No typo is found.
2. The experiments are sufficient to demonstrate the findings of previous methods’ failure in low poisoning rate scenarios. And the experiments to verify the potential reason and validate the two initial methods are also convincing.
3. The proposed methods are intuitively reasonable, which are derived step-by-step from the observations of the pre-experiments to the simple solutions, and the final version with penalty.
4. The idea is simple yet effective as shown in Sections 4 and 5, which is easy to follow.

**Weaknesses:**

In general, I appreciate this paper, but there are still some concerns from my perspective.

1. The implementation of the proposed methods is not publicly available, only the public library ‘BackdoorBench’ is provided.
2. The effectiveness of the constraint term C in equation (1) is expected to be discussed in the ablation study.

**Questions:**

1. Do the dynamic backdoor attacks, such as WaNet[1] and IAB[2], which generate triggers according to the input, follow the same phenomenon as in Figures 1 and 2? Since the WaNet is used in Section 5 while not in the pre-experiment part.

[1] Nguyen, Anh, and Anh Tran. "Wanet--imperceptible warping-based backdoor attack." *arXiv preprint arXiv:2102.10369* (2021).

[2] Nguyen, Tuan Anh, and Anh Tran. "Input-aware dynamic backdoor attack." *Advances in Neural Information Processing Systems* 33 (2020): 3454-3464.

**Limitations:**

Please see above.

---

> ### Author Rebuttal · Authors · 2023-08-09
>
> **First of all, thank you for your recognition of our work.**
>
> ### Response to Weakness 1:
>
> Thanks for your comment.
> We will release all of our code, including the corresponding tuning parameters and training checkpoints, in our final version to ensure that the results of all our experiments are reproducible.
>
> ----
>
> ### Response to Weakness 2:
>
> Thanks for this suggestion. As mentioned in Section 4, to stabilize tuning process, we add a constraint on the norm of $w$. To reduce tuning cost, we directly set it as $||w^{ori}||$ instead of manually tuning. They will be adjusted adaptively based on trained models. To show effects of our constraint, we offer the performance of FST's entire tuning process on CIFAR-10 and ResNet-18, with or without the projection term. The results are shown in **Figure 4 of one-page PDF**. The blue and purple lines represent results with or without projection. The different line type means various poisoning rate.
>
> We could clearly observe that the projection stabilizes tuning process of FST, leading to significant convergence improvement. FST quickly converges in a few epochs while achieving good robustness and clean accuracy. We will add this study to our revised version.
>
> ----
>
>
> ### Response to Question 1:
>
> Thanks for your helpful comment. Following settings in Section 3.1, we add evaluations of WaNet and IAB attacks on CIFAR-10 with ResNet-18. The results are shown in **Figure 1 of one-page PDF**.
>
> For IAB attack, Vanilla FT and LP could purify backdoored models for high poisoning rates but fail to defend against low poisoning rates attacks. WaNet can not achieve 100% ASR in low poisoning rate settings even without defense. Therefore, vanilla FT and LP could also defend against it like other defense methods shown in Section 5.2.
>
> In response to Reviewer chxU, we also extend evaluations to more models, attacks, and datasets in **Figure 3 of one-page PDF**. The results are consistent with the findings in Section 3.1.
>
> ----
>
> We hope that the above answers can address your concerns satisfactorily.

---

> > ### Comment · Reviewer_JBzP · 2023-08-17
> >
> > Thanks for your response. I am satisfied and keep my score.

---

> > > ### Author Response · Authors · 2023-08-17
> > > **Thanks!**
> > >
> > > We appreciate your further response and your recognition of our work. We will include our discussion in the revised version.
> > >
> > > The authors

---

### Official Review · Reviewer_X5wr · 2023-07-06

**Soundness:** 2 fair
**Presentation:** 3 good
**Contribution:** 4 excellent
**Rating:** 5
**Confidence:** 4

**Summary:**

This paper studies the effectiveness of finetuning in backdoor defense with a low poisoning rate and finds that the feature entanglement at low poisoning rate affects the effectiveness of finetuning. Thus, this paper proposes 3 new finetuning strategy FE-tuning, FT-init, and FST. The experiments demonstrate a promising result.

**Strengths:**

1. The finetuning study is interesting.
2. FST only uses finetuning to achieve the best backdoor removal results, which is very impressive.
3. This paper is well written and easy to follow.


**Weaknesses:**

Although this work is interesting, I still have several concerns.
1. How much data is used in FST? From my point of view, the reinitialization and the larger difference loss needs more data. But if the amount of data is too large, the defenders can use these data to retrain a new model.
2. As demonstrated in the paper, only finetuning the feature extractors cannot remove backdoor successfully. If so, it demonstrates that there is poison in the linear layers, can you explain this phenomenon?
3. Can you provide more TSNE results against more attacks like Figure3 such as BadNets and WaNet?
4. Although this paper focuses on a low poisoning rate, I think results on poisoning rate such as 20% should be also considered. Because the defenders cannot know the real poisoning rate and the robustness of different poisoning rates is important.

**Questions:**

1. How much data is used in FST? From my point of view, the reinitialization and the larger difference loss needs more data. But if the amount of data is too large, the defenders can use these data to retrain a new model.

**Limitations:**

Yes

---

> ### Author Rebuttal · Authors · 2023-08-09
>
> **First of all, thank you for your recognition of our work.**
>
> ### Response to Weakness 1 & Question 1:
>
> **Response to “How much data is used in FST?”:**
>
> As mentioned in Line 238-241 of Section 5.1, we follow previous work and only reserve either 2% or 5% of training data as the tuning dataset for both CIFAR-10 and GTSRB or CIFAR-100 and Tiny-ImageNet. The clean tuning sets consist of 1000, 768, 2500, and 5000. These subsets are relatively small in size. We will highlight details of our tuning dataset in the revised version.
>
> We also evaluate FST under a rigorous scenario with limited clean tuning samples in Section 5.3. We reduce the size of CIFAR-10 tuning set from 2% to 0.1%, which is around 50 samples. As shown in Figure 5 (c,d), FST consistently performs well across various tuning data sizes, even when the tuning set only contains 50 samples.
>
> **Response to “From my point of view, the reinitialization and the larger difference loss needs more data.”:**
>
> We think that the following points can help FST achieve a better trade-off between ACC and ASR based on very small tuning datasets:
> 1. The projection constraint $||w||_2 =C$ shrinks the range of feasible set and stabilizes the tuning process. As mentioned in Line 203-219 of Section 4, to avoid the $w$ exploding and the inner product dominating the loss, we add a constraint on $||w||$ and constrain it as $ ||w^{ori}||$ rather than manual hyperparameter. It limits feasible set on the $\ell_2$ norm ball ($ ||w^{ori}||$), which stabilizes tuning process of FST and leads to significant convergence improvement. As shown in Efficiency analysis of Section 5.3, Our method FST quickly converges in a few epochs while achieving good defense performance and clean accuracy.
> 2. when tuning on small tuning sets, the original CE loss term in our objective (Eq.(1)) help tuned model maintain clean accuracy. Compared with FE-tuning without CE loss term, FST achieves better clean accuracy and defense performance.
> 3. We only reinitialize and add difference loss to linear head. It only involves a small portion of parameters of the entire model, and feature extractor is already initialized with good accuracy.
>
> ----
>
> ### Response to Weakness 2:
>
> **Response to “As demonstrated in the paper, only ...... cannot remove backdoor successfully.”:**
>
> Actually, we do not use tuning methods that only finetune feature extractors. We speculate that the reviewer is referring to the FE-tuning. As described in Line 169 of Section 3.2 and shown in Figure 1, FE-tuning first randomly re-initialize the linear classifier and then only tune feature extractor.  We apologize for any confusion caused.  As suggested by Reviewer 2qbe, we will add detailed explanations about FE-tuning in Line 52 in our revised version.
>
> **Response to “If so, it demonstrates that there is poison in the linear layers, can you explain this phenomenon?”:**
>
> 1. FE-tuning randomly re-initializes linear classifier, so there are no poisons in new linear classifier. However, it still fails to completely eliminate backdoor triggers under various settings. The reason is that for low poisoning rate attacks, FE-tuning did not effectively shift learned features. The random initialization of linear classifier may not be sufficient to guide enough shifts on learned features. This is also why we have proposed FST with adding an extra penalty on linear classifier ($\alpha\*<w,w^{ori}>$). It encourages discrepancy between tuned and original backdoor classifier and also further promotes feature shifts, as mentioned in Section 4. The stable defense performances under various settings also verify the effectiveness of FST.
> 2. As shown in Section 3.1, under high poisoning rate, LP achieves good defense performance. This demonstrates that we could only purify backdoors in linear head to defend against attacks when backdoor features are clearly separable from clean features under high poisoning rate settings. However, as mentioned in Section 3.2, we need more feature shifts to defend against low poisoning rate attacks.
>
> ----
>
> ### Response to Weakness 3:
>
> Thanks for your suggestion. Following settings in Section 3.2, we provide TSNE results of BadNets and WaNet attacks. The results are shown in **Figure 2 of one-page PDF**.
>
> Under high poisoning rate (10%), the backdoor features (black points) are clearly separable from clean features (red points) and targeted clean and backdoor features are closer to each other under low poisoning rate (1%). This is consistent with results in Section 3.2. As shown in top row, consistent with observations of Blended attack in Section 3.2, vanilla FT still suffers from providing sufficient feature modification, leading to failed defense against BadNet attacks. WaNet fails to achieve 100% ASR in low poisoning rate settings without defense. Hence, its backdoor features are not very close to clean features, unlike Blended and BadNet attacks. Therefore, vanilla FT could also defend against it like other defense methods shown in Section 5.2.
>
> In response to Reviewer Zkrc, we also conduct visualization of adaptive attacks. As shown in **Figure 1 of one-page PDF**, FST still significantly shifts backdoor features and makes them separable from clean features.
>
>
> ----
>
> ### Response to Weakness 4:
>
> Thanks for this constructive suggestion. To fully assess FST’s effectiveness, we take more evaluations of FST by increasing poisoning rate to 30% on CIFAR-10 and GTSRB with ResNet-18. The results are shown in **Table 2 of one-page PDF**. We can observe that FST still achieves stable and outstanding defense performance under high poisoning rate, reducing ASR below 2%. This further verifies the effectiveness of our method. We will add the suggested evaluations in our revised version.
>
> ----
>
> We hope that the above answers can address your concerns satisfactorily. We would be grateful if you could re-evaluate our work based on the above responses. We look forward to receiving your further feedback.

---

> > ### Comment · Reviewer_X5wr · 2023-08-19
> >
> > Thanks for your rebuttal and most of my concerns are solved. And I will keep scores and lean to accept this paper.

---

> > > ### Author Response · Authors · 2023-08-19
> > > **Thanks for your recognition of our work**
> > >
> > > We sincerely appreciate your constructive feedback throughout the review process. We are committed to incorporating your suggestions as we revise the paper. We are delighted that our responses have addressed your concerns. Thanks for your recognition of our work!
> > >
> > > Best regards,
> > >
> > > The Authors

---

> ### Author Response · Authors · 2023-08-18
> **Seeking Your Valuable Feedback**
>
> Dear Reviewer X5wr,
>
> Thanks for spending time reviewing our work and providing valuable feedback! We have provided the response to your questions. We sincerely appreciate it if you could provide further feedback and comments on our response. Ensuring that the rebuttal aligns with your suggestions is of utmost importance. Your response is very helpful in further improving the quality of our work.
>
> Best regards,
>
> Authors

---

> > ### Comment · Area_Chair_SR9G · 2023-08-19
> >
> > Thanks to the authors for your response.
> >
> > @Reviewer X5wr: Does the rebuttal fully address your concerns?
> >
> > Best regards,
> > Your AC

---

### Official Review · Reviewer_2qbe · 2023-07-06

**Soundness:** 3 good
**Presentation:** 3 good
**Contribution:** 3 good
**Rating:** 7
**Confidence:** 4

**Summary:**

This paper observes that while fine-tuning and linear probing can act as effective defenses in the high-poisoning-rate regime, they completely fail in the low-poisoning-rate regime. The paper further shows that this is due to the fact that in the low-poisoning-rate regime, extracted features of backdoored and clean samples are highly entangled. The proposed defense is to fine-tune the backdoor model with an additional constraint to force the fine-tuned weights to differ from the original (poisoned) weights. Extensive experiments show that the proposed defense provides excellent backdoor purification while maintaining clean accuracy.

**Strengths:**

- This paper shows an interesting observation that fine-tuning and linear probing perform significantly worse as backdoor defenses in the low poisoning rate regime.
- This observation is followed by a thorough and nicely written analysis in Section 3.
- The proposed defense is simple, elegant, yet effective.
- Very extensive experiments.

**Weaknesses:**

Please pay more attention to weaknesses marked with "major" severity.
- **[minor, clarity]**: L52, please also mention that FE-tuning randomly re-initializes the classifier head. I was confused when looking at Figure 1 at first because it was not mentioned previously that FE-tuning does re-initialize $f(w)$.
- **[minor, typo]**: L102, "Here We" -> "Here, we"
- **[major, lack of experiments on high poisoning rates]**: The proposed defense should be tested with high poisoning rates just to ensure that it performs equally well.

**Questions:**

- It would be nice if the authors could extend Figure 2 to other datasets. Specifically, I am interested in seeing the results on Tiny-ImageNet and CIFAR-100.

**Limitations:**

- One potential limitation of this work is the assumption of having access to clean data for training, which may not always be practical in real-world scenarios. However, the authors have acknowledged this limitation and expressed their commitment to addressing it in future research.
- It would have been more insightful if the authors had delved deeper into why low poisoning rates cause entanglement between backdoored and clean features.

---

> ### Author Rebuttal · Authors · 2023-08-09
>
> **First of all, thank you for your recognition of our work.**
>
> ### Response to Weakness minor 1 and 2:
> Thank you for your helpful suggestion! We apologize for the confusion caused. We will add explanations about FE-tuning in Line 52 in our revised version. We will correct all the typos and carefully polish the paper in the revision.
>
> ----
> ### Response to Weakness major 1:
>
> Thanks for your valuable suggestion! To fully assess FST’s effectiveness, we take more evaluations of FST by increasing the poisoning rate to 30% on CIFAR-10 and GTSRB with ResNet-18. The results are shown in **Table 2 of one-page PDF**.
> We can observe that FST still achieves stable and outstanding defense performance under high poisoning rate setting, reducing ASR below 2%. This further verifies the effectiveness of our method. We will add the suggested evaluations in our revised version.
>
> ----
> ### Response to Questions:
>
> Thanks for your valuable suggestion. We conduct evaluations of vanilla FT and LP on CIFAR-100 and Tiny-ImageNet with pre-trained SwinTransformer. As mentioned in Section 3.1, we mainly focus on defense performance with a satisfactory clean accuracy level (80%). We tune hyperparameters based on this condition. The results are shown in **Figure 3 of one-page PDF**. We also provide results on ResNet-50, and Dense-16 on CIFAR-10 and GTSRB. We could observe that Vanilla FT and LP could purify backdoored models for high poisoning rates but fail to defend against low poisoning rates attacks. The only exception is the SSBA results since the original backdoored models have a relatively low ASR, as mentioned in Section 3.1. These results align with our findings in Section 3.1. We will add suggested evaluations in our revised version.
>
> ----
>
> ### Response to Limitation 1:
> Thanks for your comment.
> 1. We utilize a very small clean subset to conduct tuning methods in this work. As mentioned in Section 5.1, we follow previous work and only reserve either 2% or 5% of the original training data as the tuning dataset for both CIFAR-10 and GTSRB or CIFAR-100 and Tiny-ImageNet. Additionally, in Section 5.3, we also evaluate FST under a rigorous scenario with much fewer clean tuning samples. We reduce the size of our CIFAR-10 tuning set from 2% to 0.1%, which is around 50 samples. As shown in Figure 5 (c,d), FST consistently performs well across various tuning data sizes, even when the tuning set only contains 50 samples.
> 2. In real-world scenarios, we may be able to utilize existing dataset filtering methods [1] to help us construct this small tuning dataset. In this work, our main objective is to develop more robust tuning defense methods. We will investigate constructing a tuning dataset in future work.
>
> ----
>
> ### Response to Limitation 2:
>
> Thanks for your helpful suggestion! In the main submission, we mainly focus on further improving the defense performance of tuning methods against low poisoning rates attacks, after first observing that existing tuning methods fail to provide stable robustness. Here we first present some initial intuitions based on related work [2,3] and will try to investigate the reasons behind that and provide a detailed analysis in future work.
>
> Our intuition is that to learn a stable mapping between backdoor patterns in input and the target class, backdoor samples will lead the model to tend to differentiate between backdoor features and the clean features of the target class. Therefore, the former is not influenced by the latter and dominates in the model decision process, leading to the wrong classification. The high poisoning rate attacks thus lead to the obvious and easy separation in feature space. While keeping ASR, attacks with low poisoning rates lead to more stealthy backdoor features which are more close to clean features.
>
> [1]. Meta-Sift: How to Sift Out a Clean Data Subset in the Presence of Data Poisoning, Usenix 2023.
>
> [2]. Revisiting the Assumption of Latent Separability for Backdoor Defenses, ICLR 2023.
>
> [3]. Spectral signatures in backdoor attacks, NeurIPS 2018.
>
> ----
>
> We hope that the above answers can address your concerns satisfactorily.

---

> > ### Comment · Reviewer_2qbe · 2023-08-14
> > **All concerns addressed**
> >
> > Thank you for your response. Please consider adding your intuition for Limitation 2 in the revision. I am raising my rating to 7.

---

> > > ### Author Response · Authors · 2023-08-14
> > > **Thanks!**
> > >
> > > We appreciate your further comment and your recognition of our responses. Thank you for this suggestion. We will include this discussion in  Section 3 of the revised version.
> > >
> > > The authors

---

### Official Review · Reviewer_chxU · 2023-07-08

**Soundness:** 2 fair
**Presentation:** 3 good
**Contribution:** 2 fair
**Rating:** 5
**Confidence:** 4

**Summary:**

This paper finds that fine-tuning is less effective in defending against backdoor attacks with a low poisoning rate, due to the strong coupling between the clean feature and the backdoor feature. Therefore, a tuning-based backdoor purification method called feature shift tuning (FST) is proposed, which is simple and efficient. FST can effectively decouple the clean feature and the backdoor feature, and thus eliminate the backdoor in the victim model.

**Strengths:**

1. This paper proposes a simple and effective backdoor elimination method, which achieves good results in a specific scenario where the poisoning rate of the backdoor attack is relatively low.

**Weaknesses:**

1. This paper finds that fine-tuning has difficulty defending against backdoor attacks with a low poisoning rate when ResNet-18 is the victim model, but does not sufficiently demonstrate the generalizability of the problem. The same claim is not guaranteed to hold when the model capacity, model architecture, and dataset type change.

2. This paper lacks an explanation of why the method works and does not explore the effect of the range of poisoning rates on the victim model.

3. It is desirable to include backdoor elimination methods other than tuning-based methods as baseline methods as well to demonstrate the efficiency of FST.

**Questions:**

Please refer to weakness.

**Limitations:**

It is suggested that the authors provide further explanation as to why the method worked.

---

> ### Author Rebuttal · Authors · 2023-08-09
>
> **First of all, thank you for your recognition of our work.**
>
> ### Response to Weakness 1:
>
> Thanks for your constructive comments.
> We add evaluations of vanilla FT and LP using ResNet-50 (increased model capacity) and Dense-161 (different model architecture) on CIFAR-10. We also take evaluation on GTSRB (different dataset types). As mentioned in Section 3.1, we mainly focus on defense performance with a satisfactory clean accuracy level (92% on CIFAR-10, 97% on GTSRB). We tune hyperparameters based on this condition. The results are shown in **Figure 3 of one-page PDF**.
>
> We could observe that Vanilla FT and LP could purify backdoored models for high poisoning rates but fail to defend against low poisoning rates attacks. The only exception is the SSBA results since the original backdoored models have a relatively low ASR, as mentioned in Section 3.1. These results align with our findings in Section 3.1. We will add these suggested evaluations in our revised version.
>
> ----
> ### Response to Weakness 2 and Limitations:
>
> **Response to “This paper lacks an explanation of why the method works”:**
>
> 1. In lines 220-231 on page 6, we explain why FST can provide better robustness and bring unified improvements for initial methods, FE-tuning, and FT-init. Additionally, in the final figure of Figure 3, we include T-SNE visualizations of the feature extractor after applying FST to confirm its effectiveness in purifying backdoor features.
> 2. As shown in Section 3.2, we first find that in low poisoning rates scenarios, entanglements between the backdoor and clean features make FT and LP fail to defend against attacks. We propose two initial methods, FE-tuning and FT-init, to promote backdoor feature shifts and make them easily separable from the clean features of the target class. Hence, the feature extractor will no longer confuse backdoor features with clean features of the target class in the feature space. This leads that the subsequent linear classifier is difficult to be misled by backdoor samples, resulting in more robust classification. The evaluation and visualization in Section 3.2 verify their effectiveness on backdoor purification.
> 3. However, these two methods still suffer from a clean accuracy drop or unsatisfied defense performance. To achieve unified improvement, we further propose FST which actively shifts features by encouraging the difference between the tuned and original classifier. Compared with FE-tuning, FST brings an extra loss term on $\min_{w} E_{(x,y)\sim D_{T}} L(f(w; \phi(\theta;x)), y)+ \alpha <w,w^{ori}>$ to update linear classifier $w$. It further promotes feature shift by penalizing classifiers more intensely and also maintains the models’ ACC with the original CE loss term. Compared with the FT-init, by adopting $\alpha <w,w^{ori}>$, FST encourages discrepancy between tuned $w$ and original $w^{ori}$ to guide more shifts on learned backdoor features.
> 4. Feature visualization in Figure 3 shows that our FST significantly shifts backdoor features and makes them easily separable from clean features of the target class. Comparisons with other defense methods in Section 5 verify the superiority of FST in defending against attacks.
>
> **Response to “does not explore the effect of the range of poisoning rates on the victim model.”:**
>
> Thanks for this constructive comment. In the main submission, we mainly focus on attacks with low poisoning rates (0.5%, 1%, 5%) and evaluate FST on them, since Vanilla FT and LP are ineffective in defending against them in our revisiting experiments. To fully assess FST’s effectiveness, we take more evaluations of FST by increasing the poisoning rate to 30% on CIFAR-10 and GTSRB with ResNet-18. The results are shown in **Table 2 of one-page PDF**.
> We can observe that FST still achieves stable and outstanding defense performance under high poisoning rate setting, reducing ASR below 2%. This further verifies the effectiveness of our method. We will add the suggested evaluations in our revised version.
>
> ----
>
> ### Response to Weakness 3:
>
> Thanks for this valuable suggestion. To further assess FST’s effectiveness, we add another training defense method, ABL [1], which also performs well in BackdoorBench. Following settings of Section 5.1, we evaluate it in CIFAR-10 with ResNet-18. The results are shown in the below table. We could observe that FST achieves better and more stable performance than ABL under various settings with maintaining good clean accuracy.
>
> [1] Anti-backdoor learning: Training clean models on poisoned data, NeurIPS 2021.
>
> **CIFAR-10 and ResNet-18**
> |Attack|Poisoning rate|ABL(ACC/ASR)|FST(ACC/ASR)|
> |:-------:|:-------:|:-------:|:-------:|
> |BadNets|5%|89.65/0.08|93.17/0.00|
> ||1%|72.34/7.13|92.81/0.01|
> ||0.50%|71.59/9.97|93.63/0.02|
> |Blended|5%|86.36/0.74|92.87/3.07|
> ||1%|71.48/18.77|93.59/0.19|
> ||0.50%|71.82/49.66|93.15/0.06|
> |WaNet|5%|69.78/77.39|91.56/0.26|
> ||1%|72.08/31.17|91.83/0.51|
> ||0.50%|74.31/13.23|91.70/0.78|
> |SSBA|5%|73.44/24.61|93.48/0.27|
> ||1%|73.72/72.57|93.32/0.56|
> ||0.50%|73.33/26.29|92.97/0.04|
> |SIG|5%|89.31/2.91|93.24/0.02|
> ||1%|89.73/5.37|93.09/0.03|
> ||0.50%|74.07/50.60|93.18/0.01|
> |LC|5%|90.19/4.29|93.47/0.68|
> ||1%|67.41/8.83|93.51/0.30|
> ||0.50%|75.42/99.84|93.44/1.71|
>
> ----
>
> We hope that the above answers can address your concerns satisfactorily. We would be grateful if you could re-evaluate our work based on the above responses. We look forward to receiving your further feedback.

---

> ### Author Response · Authors · 2023-08-18
> **Seeking Your Valuable Feedback**
>
> Dear Reviewer chxU,
>
> Thanks for spending time reviewing our work and providing valuable feedback! We have provided the response to your questions. We sincerely appreciate it if you could provide further feedback and comments on our response. Ensuring that the rebuttal aligns with your suggestions is of utmost importance. Your response is very helpful in further improving the quality of our work.
>
> Best regards,
>
> Authors

---

> > ### Comment · Area_Chair_SR9G · 2023-08-19
> >
> > Thanks to the authors for your response.
> >
> > @Reviewer chxU: Does the rebuttal fully address your concerns?
> >
> > Best regards,
> > Your AC

---

> > ### Author Response · Authors · 2023-08-21
> > **Thank you**
> >
> > Dear Reviewer chxU,
> >
> > We want to follow up to make sure that we can discuss any further questions/comments/concerns. Please let us know if we could do anything to resolve any questions or concerns between now and the end of the discussion period. The discussion phase is coming to an end.
> >
> > The authors

---

### Official Review · Reviewer_Zkrc · 2023-07-08

**Soundness:** 3 good
**Presentation:** 3 good
**Contribution:** 3 good
**Rating:** 5
**Confidence:** 4

**Summary:**

This paper proposes a novel backdoor defense approach called Feature Shift Tuning (FST) that actively promotes feature shifts to disentangle the learned features of a deep neural network. The authors conduct a thorough assessment of widely used tuning methods, vanilla Fine-tuning (FT), and simple Linear Probing (LP), and demonstrate that they both completely fail to defend against backdoor attacks with low poisoning rates.

FST is an end-to-end tuning method that actively shifts features during fine-tuning by encouraging the difference between the tuned classifier weight w and the original backdoored classifier weight w_ori. FST significantly shifts backdoor features and makes them easily separable from clean features of the target class, which effectively mitigates backdoor attacks. In summary, the proposed FST method is a strong backdoor defense paradigm that can achieve stable improvements in both robustness and clean accuracy compared to other initial methods such as FE-tuning and FT-init.


**Strengths:**

- The proposed FST (Fine-tuning with Separated Transformations) method is a strong backdoor defense paradigm that can achieve stable improvements in both robustness and clean accuracy compared to other initial methods such as FE-tuning and FT-init. FST is an end-to-end tuning method that actively shifts features during fine-tuning by encouraging the difference between the tuned classifier weight w and the original backdoored classifier weight w_ori. This is giving a good reference to the research on this aspect.

- FST disentangles the clean and backdoor features, which helps to purify the model. It significantly shifts backdoor features and makes them easily separable from clean features of the target class, which effectively mitigates backdoor attacks. FST outperforms other methods by a large margin for backdoor robustness on CIFAR-10 and GTSRB datasets.

- FST is effective against adaptive attacks such as the Adaptive-Blend attack, achieving an average drop on ASR by 81.66% and 91.75% respectively for the CIFAR-10 and GTSRB datasets. FST is a unified method that addresses both the clean accuracy drop and unsatisfied robustness improvements that are present in other backdoor defense methods. The ablation study is organized well to clearly demonstrate the whole proposed method. And it makes the paper easy to follow.


**Weaknesses:**

- [Contribution and novelty] The contribution of paper is somehow incremental, as the method is developed based on previously researched findings. The backdoor attack can be erased by decoupling the feature extractor and the linear classifier in [1].

- [Adaptive Attack] There, the submission only briefly mentions an attempt at achievingthe  disentanglement between the clean and backdoor features through an adaptive attack, but fails to do so effectively, and provides only limited insight why the adaptive attack failed. Please take a closer look at providing a strong evaluation of the adaptive attack scenario. Besides, the paper [2] also proposes another defense, Adaptive-Patch, which should also be discussed when attacking the propsoed FST.

[1] Huang, Kunzhe, et al. "Backdoor Defense via Decoupling the Training Process." International Conference on Learning Representations. 2021.

[2] Xiangyu Qi, Tinghao Xie, Yiming Li, Saeed Mahloujifar, and Prateek Mittal. Revisiting the assump405
tion of latent separability for backdoor defenses. International Conference on Learning Representations. 2023.

**Questions:**

Listed in the weakness of the paper. Score can be improved if concerns listed above are resolved.

**Limitations:**

The authors addressed the limitation that they assume the defender would hold a clean tuning set which may not be feasible in certain scenarios.

---

> ### Author Rebuttal · Authors · 2023-08-09
>
> **First of all, thank you for your recognition of our work.**
>
> ### Response to Weakness 1:
>
> **Response to “The contribution of paper ...... previously researched findings.”:**
>
> 1. The authors of [1] propose that end-to-end supervised training makes the model learn backdoor features. Hence, they propose a two-phase training defense method called DBD. DBD utilizes self-supervised learning (SSL) to train feature extractor and then trains a linear classifier with the frozen feature extractor.
> 2. In contrast to DBD, our work focuses on how to efficiently purify existing backdoor triggers during tuning process, rather than training process. We first investigate if common FT methods in the pretrain-tuning paradigm, such as vanilla FT and LP, can consistently provide robustness against attacks across various settings. **We find that**: 1) under high poisoning rate setting where backdoor and clean features of targeted class are well separable, simple LP could provide satisfactory defense performance; 2) under low poisoning rate setting, backdoor and clean features are tangled together (Figure 2 of Section 3.1). As a result, FT and LP fail to purify inserted backdoors due to insufficient feature shifts.
> While DBD shows that feature extractor can learn backdoor features, it does not discuss how learned features vary at different poisoning rates, particularly in terms of separability between clean and backdoor features. This is crucial for designing robust and stable defense methods for various attack settings.
> 3. Motivated by our findings, we propose two initial methods (FE-tuning and FT-init) to encourage separability between backdoor and clean features. However, they still suffer from a clean accuracy drop or unsatisfied robustness improvements. To achieve unified improvement, we further propose FST based on them, a simple end-to-end tuning method. It actively encourages discrepancy between tuned and original classifiers to guide more shifts on backdoor features.
>
> **Response to “The backdoor attack can ...... and the linear classifier in [1].”:**
>
> 1.  As mentioned above, FST is an end-to-end tuning method, which does not decouple the feature extractor and linear classifier like [1]. Our initial method, FE-tuning, adopts a decoupling form in tuning process. However, they still suffer from a clean accuracy drop or unsatisfied robustness improvements (Table 1 of Section 3.2). As shown in Section 5.2, FST achieves better and more stable defense performance under various settings. Compared with high training costs of DBD, FST is a much simpler and more flexible method that can be easily integrated with existing training methods and pretrained models.
>
> ----
>
> ### Response to Weakness 2:
>
> Thanks for your suggestion. We first give reasons why we believe attacks [2] could be powerful adaptive attacks against our method. Then, we follow your suggestion and conduct a more detailed adaptive attack evaluation based on [2]. We finally provide explanations for why adaptive attacks failed.
> 1. As shown in Section 3.1, under high poisoning rate, backdoor features are clearly separable from clean features of targeted class and thus could be easily purified by vanilla FT and LP. To bypass defense methods based on latent separation property, Adaptive poisoning attacks [2] actively suppress the latent separation between backdoor and clean features. So that they can achieve a high ASR by adopting extremely low poisoning rate and adding regularization samples. This also corresponds to our finding in Section 3.2 that entanglements between clean and backdoor features in low poisoning rate settings make FT and LP fail. The evaluations in [2] also show that these attacks successfully bypass existing strong latent separation based defense. Hence, we believe it is also equally a powerful adaptive attack against our FST method.
> 2. Following the reviewer’s suggestion, we add evaluations of Adaptive-patch attack on CIFAR-10 and ResNet-18. To further reduce latent separability and improve adaptiveness against latent separation based defenses, we also use more regularization samples, following ablation study of Section 6.3 [2]. We show results in **Table 1 of one-page PDF**. We can observe that even against more stealthy adaptive attacks, FST still achieves outstanding defense performance.
> 3. Our work and [2] mainly focus on practical data-poisoning backdoors. To further assess stability of FST, we also test FST against training-control adaptive attacks [3]. The authors [3] utilize an adversarial network regularization during training process to minimize differences between backdoor and clean features in latent representations. Since the authors do not provide source code, we implement it based on descriptions in original paper. The results are shown in **Table 1 of one-page PDF**. FST still significantly reduces ASR. This further proves the excellent stability of our method.
> 4. To explain why adaptive attacks fail, we provide TSNE visualizations of learned features from backdoored models, as shown in Section 3.2. We show results in **Figure 1 of one-page PDF**. We can first observe that adaptive attacks significantly reduce latent separability. Clean and backdoor features are tightly tangled. FST effectively shifts backdoor features and makes them easily separable from clean features of the target class. Therefore, the feature extractor will no longer confuse backdoor features with clean features of target class in feature space. This leads that subsequent simple linear classifier is difficult to be misled by backdoor samples, resulting in more robust classification.
>
> We will include adaptive attack evaluations in the revision.
>
> [3]. Bypassing backdoor detection algorithms in deep learning, Euro S&P.
>
> ----
>
> We hope that our answers can address your concerns satisfactorily and improve the clarity of our contribution. We would be grateful if you could re-evaluate our paper. We look forward to receiving your further feedback.

---

> > ### Comment · Reviewer_Zkrc · 2023-08-18
> >
> > Thanks for the authors' detailed response. This work presents convincing results but does not surprise me due to the novelty. Given all, I maintain my score.

---

> > > ### Author Response · Authors · 2023-08-18
> > > **Thanks for your response**
> > >
> > > Thanks for your recognition of our work and feedback on our response. We will include your valuable suggestions and our discussion in the revised version.
> > >
> > > We will further clarify our contributions and the technical novelty in the revised version.
> > >
> > > Thanks
> > >
> > > The authors

---

### Author Rebuttal · Authors · 2023-08-09

Dear Reviewers,

Thank you for your recognition of our work! We sincerely appreciate all of your precious time and constructive comments.

All these comments and suggestions are very insightful and beneficial for us to improve the quality of this work. We have responded to each review separately and hope our responses are helpful in addressing the reviewers' questions. We will carefully revise our manuscript by adding suggested evaluations, providing more detailed explanations, and fixing the typos.

**we have attached a separate PDF file containing additional experimental results aimed at addressing the concerns of reviewers.**

Best regards,

The Authors

---

> ### Author Response · Authors · 2023-08-19
>
> Thanks to reviewers. Please let us know if we could do anything to resolve any questions or concerns between now and the end of the discussion period. We are eagerly looking forward to receiving your valuable feedback and comments on the points we addressed in the rebuttal. Ensuring that the rebuttal aligns with your suggestions is of utmost importance. Hope you have a nice weekend!
>
> The Authors

---

### Decision · Program_Chairs · 2023-09-21

**Decision:**

Accept (poster)

**Comment:**

The reviewers agree to accept the paper, with 3 Borderline Accept and 2 Accept. Overall, the paper provides an important discovery and a simple but effective backdoor defense method. The ACs agree with the reviewers' decision.